# Optical-Tweezers-integrating-Differential-Dynamic-Microscopy maps the spatiotemporal propagation of nonlinear strains in polymer blends and composites

**Karthik R. Peddireddy** [1], **Ryan Clairmont**[1], **Philip Neill**[1], **Ryan McGorty** [1] & **Rae M. Robertson-Anderson** [1] ✉

How local stresses propagate through polymeric fluids, and, more generally, how macromolecular dynamics give rise to viscoelasticity are open questions vital to wide-ranging scientific and industrial fields. Here, to unambiguously connect polymer dynamics to force response, and map the deformation fields that arise in macromolecular materials, we present Optical-Tweezers-integrating-Differential -Dynamic-Microscopy (OpTiDMM) that simultaneously imposes local strains, measures resistive forces, and analyzes the motion of the surrounding polymers. Our measurements with blends of ring and linear polymers (DNA) and their composites with stiff polymers (microtubules) uncover an unexpected resonant response, in which strain alignment, superdiffusivity, and elasticity are maximized when the strain rate is comparable to the entanglement rate. Microtubules suppress this resonance, while substantially increasing elastic storage, due to varying degrees to which the polymers buildup, stretch and flow along the strain path, and configurationally relax induced stress. More broadly, the rich multi-scale coupling of mechanics and dynamics afforded by OpTiDDM, empowers its interdisciplinary use to elucidate non-trivial phenomena that sculpt stress propagation dynamics–critical to commercial applications and cell mechanics alike.

Entangled polymeric fluids, blends and composites–ubiquitous in the natural and commercial world–exhibit complex viscoelastic behavior[1–5], with the characteristics of stress dissipation and storage dictated by the dynamics and relaxation mechanisms of the comprising polymers, as well as inter-polymer interactions and entanglements. The relative viscous dissipation versus elastic storage of induced stress, and the associated spatiotemporal scales, rely on how polymers move, deform, rearrange, stretch, and distribute stress to neighboring polymers.

Despite this critical coupling of polymer dynamics with material stress response and deformation, current techniques to probe entangled polymers, as well as other complex fluids and soft materials, are primarily designed to measure either force response or macromolecular transport and dynamics. For example, bulk rheology measures the viscoelastic properties and stress response of the bulk fluid in response to macroscopic shearing. On the other hand, typical microrheology measurements track the Brownian fluctuations of individual microscale probes embedded in the material to extract viscoelastic properties using the generalized Stokes-Einstein relation (GSER)[6]. Variations to the GSER approach have employed dynamic light scattering (DLS)[7,8] and differential dynamic microscopy (DDM)[9,10] to extract viscoelastic properties from Brownian fluctuations. These microscale methods require <10 μL of material and are able to resolve spatial heterogeneities inaccessible to bulk rheology; however, the

[1]Department of Physics and Biophysics, University of San Diego, San Diego, CA 92110, USA. ✉e-mail: randerson@sandiego.edu

passive nature of the measurements restricts applicability to the linear regime. Moreover, choosing an appropriate probe size and surface treatment for a given system is both non-trivial and essential for correctly interpreting the data[11–13].

Active microrheology methods employing, e.g., magnetic or optical tweezers, are able to apply stress or strain in a local region of the fluid and measure the force the polymers exert to resist the strain[14–16]. For example, using optical tweezers (OpT) microrheology a trapped microscale probe can be pulled through the fluid with precisely controlled rates and distances such that the force response in the nonlinear regime (in which the fluid is pushed far from equilibrium by fast and/or large strains) and at mesoscopic scales (10's of microns) can be measured[17]. Recent advances to OpT microrheology include, for example, coupling OpT to microfluidics to enable force measurements during in-situ modulation of environmental conditions[18,19]; and integrating multiplane microscopy into OpT to allow for video tracking of a trapped bead in three dimensions to improve accuracy of microrheological properties[20].

To complement these rheological measurement techniques, macromolecular dynamics and transport can be measured by tracking the Brownian fluctuations of single fluorescent-labeled molecules using particle-tracking algorithms[15,21–23]. DLS[7,8] and DDM[9,10,24] can also characterize dynamics and transport by examining ensemble-averaged macromolecular fluctuations. DDM is particularly advantageous for certain systems as it can probe larger spatiotemporal scales than particle-tracking, and can extract information from weaker signals in which single molecules cannot be resolved[9,10,24,25]. The deleterious effects of photobleaching are also comparatively reduced as measurement accuracy does not rely on tracking single polymers over extended time periods. DLS has similar advantages as DDM but does not use microscope images as its input so cannot be coupled to other microscopy techniques such as OpT in a straightforward manner.

In efforts to couple stress response to macromolecular dynamics, opto-rheometers have been used to correlate bulk rheological response with macromolecular deformation, and microfluidic shearing devices have been used to measure polymer stretch and relaxation under extensional flow[26–33]. These methods have shed important light on the dynamics of single polymers under uniform strain and flow, revealing, e.g., shear banding and tumbling of entangled linear DNA under bulk shear[34], and heterogeneous relaxation and swelling of ring DNA under extensional micro-flows due to transient threading by linear chains[35]. However, these techniques require relatively large sample volumes (~$10^2$–$10^3$ μL) which limit their use to study highly entangled biopolymer solutions and other valuable or liable biomaterials which are time and resource intensive to prepare. Moreover, these methods are not equipped to measure scale-dependent, hierarchical phenomena that often emerge in polymer mixtures, or map how stress from a local disturbance is distributed and propagated through the network via the strain, displacement, and deformation of the surrounding polymers.

The complexity of available relaxation mechanisms, transport modes, inter-polymer interactions, and conformational dynamics of entangled polymers–greatly amplified for blends and composites of polymers with different sizes, stiffnesses and topologies[22,35–46]–demands techniques that can unequivocally correlate polymer transport and dynamics to force response and relaxation, and elucidate the spatiotemporal distribution and propagation of deformation and strain. For example, in blends of ring and linear polymers, threading of rings by linear chains has been shown to lead to emergent enhanced viscosity and shear-thinning, along with longer relaxation timescales and suppressed terminal regime scaling[38,47–52]. Threaded rings have been shown to both swell and collapse depending on the sizes of the rings and linear chains and undergo more heterogeneous transport than their linear counterparts[53]. At the same time, ring-ring entanglements are weaker and less persistent, leading to faster relaxation and reduced shear-

thinning, owing to their reduced ability to retain entanglements and entropically stretch in the shear direction[50,54]. Although these diverse phenomena suggest rich and complex modes of strain propagation and redistribution, how polymers in ring-linear blends deform in response to stress, and the extent to which the deformation impacts neighboring entangled polymers remains completely unexplored.

Polymer topology also plays an important role in the force response of composites of flexible and stiff polymers. For example, previous studies on composites of DNA and rigid microtubules (MT) showed that composites comprising linear DNA undergo microscale phase separation and MT flocculation; however, exchanging linear DNA for rings of the same size and concentration promotes DNA-MT mixing and hinders MT polymerization[45]. These highly distinct architectures give rise to a dramatic non-monotonic dependence of force on MT concentration for composites with linear DNA, with the resistive force first increasing then decreasing as MT concentration is increased, due to increasing MT flocculation that reduces the connectivity and percolation of the network. Conversely, DNA-MT composites formed with ring DNA exhibit much smaller and monotonic increase in force with increasing MT concentration. These findings suggest that DNA-MT composites composed of comparable concentrations of ring and linear DNA may be most effective at maintaining MT connectivity and conferring high strength and resistance to deformation. The polymer dynamics that lead to this hypothesized behavior have yet to be explored.

Because MTs are $10^4 \times$ stiffer than DNA ($l_{p,MT} \simeq 1$ mm vs $l_{p,DNA} \simeq$ 50 nm)[55–57] and ~$10 \times$ thicker (~25 nm vs 2 nm)[55,58], their response to induced stress, and the spatiotemporal scales over which they can deform to dissipate and distribute stress, are widely different than those for DNA. For example, DNA polymers typically assume random coil configurations in steady-state, as opposed to the extended profile of rigid rod MTs. Further, DNA under shear can entropically stretch, bend, dis-entangle, and reorient to dissipate stress, while MTs are much more limited in their dynamical response[39,43,59,60]. Timescales associated with configurational relaxation, e.g., the disengagement time $\tau_D$ over which a polymer reptates out of its entanglement tube, are orders of magnitude longer for MTs compared to DNA[11,46,55,61,62], such that entangled MT solutions respond largely elastically, with minimal stress relaxation, while entangled DNA displays viscoelasticity with terminal viscous flow behavior in response to strains that are slower than $\tau_D$.

Here, we introduce an experimental technique–Optical Tweezers integrating Differential Dynamic Microscopy (OpTiDDM)–to directly measure the macromolecular deformations and dynamics induced by local linear and nonlinear disturbances and map the associated propagation field. Specifically, we apply continuous cycles of nonlinear straining in a local region of a polymer network while simultaneously imaging single fluorescent-labeled polymers (DNA) surrounding the strain and measuring the force the network exerts on the probe. We demonstrate unambiguous coupling between macromolecular dynamics and resistive forces in response to strain to reveal non-trivial relationships between the stress response, relaxation, macromolecular strain alignment, propagation, and flow. Our measurements with blends of ring and linear DNA and their composites with stiff microtubules demonstrate the power of OpTiDDM to discover unexpected physical phenomena and dissect complex and subtle relationships between various metrics of dynamics and mechanics. For example, we show that while resistive forces increase monotonically with increasing strain rate for all networks, macromolecular deformation dynamics and propagation show a distinct non-monotonic rate dependence that depends on the structural composition of the network. Moreover, while DNA-MT composites exhibit the strongest elastic-like force response, highly entangled DNA blends exhibit the most pronounced propagation field and scale-dependent dynamics.

## Results

### OpTiDDM couples macromolecular dynamics to stress response and propagation

Because this work introduces OpTiDDM, with a primary aim of enabling its use by other researchers, we first describe the critical components and rationale for the technical design, which is based on a force-measuring optical tweezers (OpT) outfitted with an epi-fluorescence microscope (Fig. 1). In our setup, an optically trapped microsphere probe (Fig. 1a) can be translated through a sample at precise distances $s$ (up to 50 μm) and rates $\dot{\gamma}$ (up to 100's of μm/s) using a piezoelectric mirror to move the trap or a piezoelectric stage to move the sample relative to the trap[17]. Simultaneously, we can measure the force the sample exerts on the probe by using a position-sensing detector (PSD) to measure the laser deflection, and image the surrounding polymers using the fluorescence microscopy capability. These features allow us to impart nonlinear and mesoscale strains, measure the resulting local stress response (Fig. 1b), and perform DDM analysis to extract macromolecular dynamics and deformations (Fig. 1c).

For force measurements, we impart strains by moving the stage (keeping the trap fixed) to ensure that the laser deflection is entirely from the force imparted on the probe and not from the moving trap (Fig. 1b)[11,17,36,38,42,43,61–66]. To determine the impact of cyclic straining on the polymer dynamics, and map the strain-induced deformation field, we impart strains by moving the trap (keeping the sample fixed) to allow for precise imaging of the thermal motion and strain-induced deformations of polymers in the field-of-view (FOV). Specifically, we use the fluorescence capability of our OpT-enabled microscope to image fluorescent-labeled DNA tracer molecules embedded in the sample and filling a 78 μm × 117 μm FOV centered on the strain path of the probe (Fig. 1c), and record time-series of the moving DNA tracers throughout the duration of the straining. We then perform DDM on spatially resolved regions of interest (ROI), each (16.6 μm)², centered horizontally (along $x$) with the strain path and vertically (orthogonal to the strain) at distances of $y = 8$ μm $\simeq s/2$ to $y = 27$ μm $\simeq 2s$ from the strain path (Fig. 1c).

We use DDM instead of more conventional single-particle-tracking (SPT) and particle image velocimetry (PIV) methods to allow for a high density of labeled macromolecules that are smaller, dimmer, and more susceptible to photobleaching as compared to microspheres and other standard probes[9,10]. The high density, which prevents localizing and tracking single particles, is critical to spatially resolving statistically robust dynamics within each ROI. Lower signal threshold than is needed for SPT and PIV facilitates using DNA or other macromolecules, which can also stretch and align with the strain, further complicating tracking methods on which PIV and SPT rely. Although implementations of DDM often use microsphere tracers to probe network dynamics, we use the comprising polymers themselves to directly visualize and report the macromolecular dynamics (Fig. 1c). We have previously established and validated the use of DDM for characterizing both passive and active transport of DNA and other biopolymers in crowded and entangled environments[22,23,33,40,41,67,68].

Our strain program, which we designed to accurately measure polymer deformation and relaxation in response to nonlinear straining, and optimize statistical significance and signal-to-noise, consists of cyclically sweeping an optically trapped probe forward and backward through a horizontal distance of $s = 15$ μm at controlled strain rates $\dot{\gamma}$ that span a ~20-fold dynamic range ($\dot{\gamma} = 9.4$–$189 \, \text{s}^{-1}$). The maximum and minimum rates are determined by the strength of the optical trap and the requirement of >5 oscillations (to ensure statistical significance) before noticeable photobleaching occurs, respectively. Each oscillatory shear persists for 50 s, including cessation periods of $\triangle t_R = 3$ s between each sweep to allow the network to relax (Fig. 1b and SI Fig. S2), such that

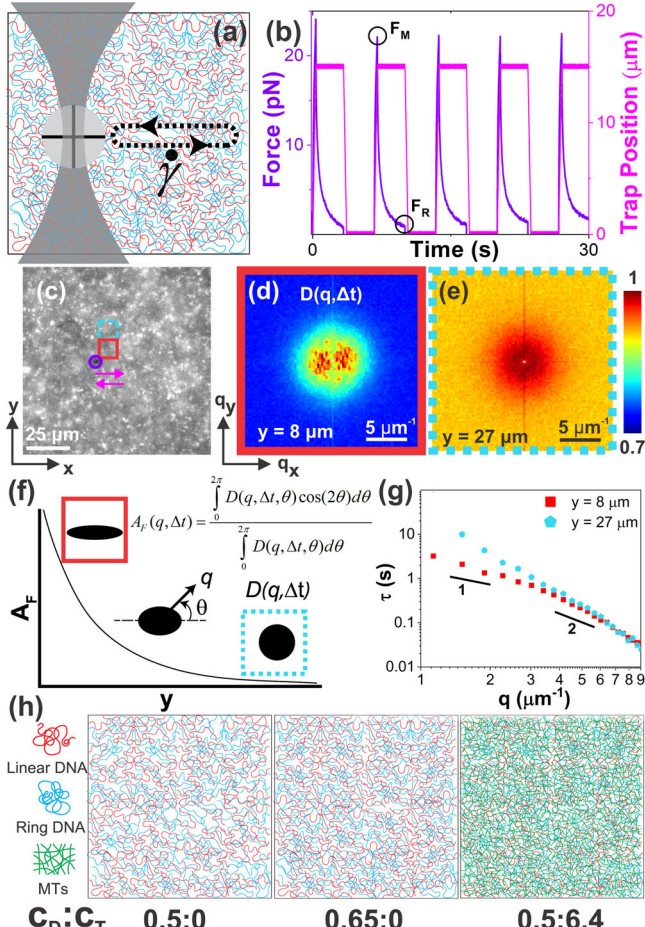

**Fig. 1 | OpTiDDM couples stress response to macromolecular dynamics and their propagation. a, b** Optical tweezers (OpT): **a** A microsphere probe (gray circle) embedded in a polymer network is trapped using a focused laser beam, and cyclically swept through a distance $s = 15$ μm at strain rates of $\dot{\gamma} = 9.4$–$189 \, \text{s}^{-1}$ for 50 s, including $\Delta t_R = 3$ s cessation periods between each sweep. **b** We measure the trap position (magenta) and force exerted on the probe (purple), and evaluate the maximum force reached during each sweep $F_M$ and the residual force at the end of each cessation period $F_R$. **c** During each strain, we collect time-series of fluorescent-labeled DNA molecules in the networks. The purple circle and magenta arrows indicate the probe position and direction of motion. We divide the FOV into (16.6 μm)² ROIs centered horizontally with the strain path at 10 vertical distances centered at $y = \pm 8$ μm (red border shows +8 μm) to $y = \pm 27$ μm (blue-dashed border shows +27 μm) above (+) and below (−) the strain path, and determine the corresponding DDM image structure function $D(\mathbf{q}, \Delta t)$. **d, e** $D(q_x, q_y)$ plots at a lag time $\Delta t = 0.25$ s for boxed-in ROIs show that **d** near the strain ($y = 8$ μm), $D(\mathbf{q}, \Delta t)$ is anisotropic with two lobes that lie along the strain axis, while **e** far from the strain ($y = 27$ μm), $D(\mathbf{q}, \Delta t)$ is radially symmetric. Colorscale indicates the magnitude of $D(\mathbf{q}, \Delta t)$, normalized by the corresponding maximum $D(\mathbf{q}, \Delta t)$ value. **f** The alignment factor, $A_F(y)$, schematically portrayed and computed using the equation shown, quantifies the preferential alignment of $D(\mathbf{q}, \Delta t)$ with the strain path, generally decreasing with increasing $y$ as qualitatively depicted. **g** DDM decay times $\tau(q)$ determined by fitting $D(q, \Delta t)$ describe the macromolecular dynamics, with diffusive and ballistic dynamics described by $\tau(q) \sim q^{-2}$ and $\tau(q) \sim q^{-1}$ respectively. **h** Cartoons of the polymer networks we investigate, defined by the mass concentration (mg/ml) of the ring-linear DNA blend $c_D$ (0.5 or 0.65) and molar concentration (μM) of tubulin dimers $c_T$ (0 or 6.4) (see Fig S1). Data shown in **b–g** is for the 0.5:0 system subject to a strain rate of $\dot{\gamma} = 42 \, \text{s}^{-1}$.

each independent measurement includes 6–9 strain cycles (depending on $\dot{\gamma}$; Fig. 1).

We focus our DDM analysis on three metrics that can be extracted from the videos by analyzing their image differences in Fourier space and computing an image structure function $D(\mathbf{q}, \triangle t)$ for varying

wavevectors $q$ of each pair of image differences separated by a lag time $\triangle t$ (see Fig. 1c–g and SI Methods section)[9,10,24,69]. The first metric we examine is the alignment factor $A_F$ (see Fig. 1f and SI Methods section)[70], to determine the degree to which polymer motion is preferentially aligned along the strain direction versus randomly distributed. Increasing fractional values of $A_F$ correlate with increasing alignment, whereas $A_F = 0$ indicates completely isotropic motion.

The second metric is the density fluctuation decay time $\tau(q)$ (Fig. 1g), determined by fitting the intermediate scattering function (ISF) determined from $D(q, \triangle t)$, which describes the type of motion, e.g., ballistic, superdiffusive, diffusive, subdiffusive, etc.[9,10] Specifically, the scaling exponent associated with the power-law dependence of $\tau(q)$, i.e., $\tau(q) \sim q^{-\beta}$, provides this characterization, with $\beta = 1$, $\beta = 2$, $\beta > 2$ and $1 < \beta < 2$ corresponding to ballistic, diffusive, subdiffusive, and superdiffusive dynamics, respectively.

The third metric is the exponent $\delta$ that stretches or compresses the exponential function that describes the ISF (see SI Methods section). Dilute and isotropic thermal systems undergoing diffusive (Brownian) motion exhibit $\delta = 1$ scaling[9,71]. However, superdiffusive or ballistic-like motion, as seen in actively driven systems, often manifest $\delta > 1$[72–74], and subdiffusive dynamics that crowded and heterogeneous systems exhibit are typically better fit to $\delta < 1$[23,25,68,71,75].

Importantly, our implementation of DDM in OpTiDDM enables precise quantification of macromolecular dynamics over 3 decades of length and time scales (~0.6 μm – 30 μm, 20 ms – 100 s), well above and below the range that typical implementations of SPT and PIV can measure[23,69], particularly for dense or noisy systems. Moreover, our dynamical characterization of the surrounding network–which quantifies macromolecular dynamics, deformations and relaxation profiles in response to strain–is distinct and arguably much richer than mapping a simple strain or displacement field that PIV can generate. While PIV can determine the displacement averaged over a portion of the imaged FOV and over a given lag time, it is relatively insensitive to the random thermal motion of molecules. However, how macromolecules respond to random thermal forces (e.g., whether and to what extent the diffusive motion is anomalous) can provide important insight into the nature of crowded and entangled systems. Therefore, we use DDM to not only determine the degree to which the motion of the macromolecules align with the strain, but to also quantify the random thermal fluctuations.

To demonstrate the efficacy, sensitivity, and wide applicability of OpTiDDM, we perform measurements on entangled solutions of blended ring and linear DNA molecules with equal contour lengths of $L = 115$ kbp $\simeq 39$ μm, as well as their composites with microtubules (MT) (Fig. 1h). In all three systems that we investigate, the ring:linear DNA ratio is fixed to 1:1, a ratio which has been shown to give rise to enhanced elastic strength and shear-thinning compared to the corresponding single-component systems[38,52,76], as well as maximal propensity for threading of rings by linear chains[38,44]. We examine blend concentrations of 0.5 mg/ml (~$3c_e$ where $c_e$ is the critical entanglement concentration[77]) and 0.65 mg/ml (~$4c_e$), as well as a composite of the 0.5 mg/ml DNA blend and MTs polymerized from 6.4 μM tubulin. The MT concentration was chosen such that the corresponding entanglement tube diameter $d_T$ is comparable to that of the 0.65 mg/ml DNA solution (see SI Methods section) such that we can isolate the dependence of polymer stiffness on the phenomena we discover. Throughout the paper, we refer to these three systems by their corresponding DNA mass concentration $c_D$ (in mg/ml) and tubulin molar concentration $c_T$ (in μM), i.e., $c_D : c_T$ = 0.5:0, 0.65:0, and 0.5:6.4. Importantly, these systems are only subtly different from one another, with <25% DNA concentration differences, minimal changes in entanglement spacing and constant length and topologies of the DNA. In this way, we are able to demonstrate the sensitivity of OpTiDDM to measure variations in the dynamics and stress response of networks that are minimally distinct.

## Strain alignment of entangled DNA exhibits non-monotonic dependence on strain rate

We first examine the degree to which the motion of DNA polymers surrounding the local strain deviates from isotropic thermal fluctuations and preferentially aligns with the strain path (Fig. 1d, e). We observe that $D(q_x, q_y)$ is anisotropic close to the strain, with two lobes that lie along the strain path ($x$-axis), indicating preferred strain-aligned motion. Conversely, $D(q_x, q_y)$ is isotropic and radially symmetric far from the strain path. The reduced variation in $D(q_x, q_y)$ values far from the strain (Fig. 1e), compared to close to the strain (Fig. 1d) indicates slower dynamics (less decorrelation).

To quantify the $y$-dependent strain alignment, we compute the alignment factor $A_F(y)$ (Figs. 1f and 2a). For an intermediate strain rate ($\dot{\gamma} = 42\,s^{-1}$), we find that $A_F$ decreases to zero as $y$ increases for all three systems. Closest to the strain, the high-concentration DNA blend (0.65:0) exhibits the strongest alignment (largest $A_F$) of all 3 systems, while it exhibits the weakest propagation of this alignment to further $y$ distances from the strain. Moreover, this effect disappears for lower or higher strain rates (Fig. 2b), in which the 0.65:0 blend actually displays the weakest alignment (lowest $A_F$) among the three systems near the strain but the strongest alignment propagation. Another unexpected feature of Fig. 2a, b is that $A_F(y)$ for both high (189 $s^{-1}$) and low (9.4 $s^{-1}$) strain rates are smaller than those for $\dot{\gamma} = 42\,s^{-1}$. In other words, faster strains do not lead to stronger strain alignment. This non-monotonic dependence of $A_F(y)$ on $\dot{\gamma}$, clearly shown in Fig. 2c, is strongest for the 0.65:0 blend and weakest for the DNA-MT composite (0.5:6.4), resulting in the DNA-MT composite exhibiting stronger strain alignment at low and high $\dot{\gamma}$, compared to the 0.65:0 blend.

To shed further light on this intriguing behavior, we more closely examine the $\dot{\gamma}$ dependence of the alignment for each system independently (Fig. 2c). $A_F(y)$ for the 0.5:0 blend displays a non-monotonic $\dot{\gamma}$ dependence, with both the strength and propagation of $A_F$ maximized at $\dot{\gamma} = 42\,s^{-1}$. The 0.65:0 blend exhibits similar non-monotonic dependence near the strain path, but at larger distances ($y > 15$ μm), the slowest strain rate actually exhibits the strongest alignment. This strong low-$\dot{\gamma}$ alignment propagation is also evident in the DNA-MT composite (0.5:6.4), suggesting that robust connectivity, lacking in the more weakly entangled 0.5:0 system, is required for this phenomenon. We understand this effect as follows: The slow rate minimizes potential strain-induced dis-entangling, de-threading, and shear-thinning–all of which reduce network connectivity–such that the entangled networks remain more connected at lower strain rates thereby facilitating alignment. Conversely, polymers comprising the 0.5:0 system, which have fewer entanglements, can more easily deform and move to relax induced stress, thereby reducing the propagation of stress to surrounding polymers at low strain rates.

Comparing the DNA-MT composite (0.5:6.4) to the pure DNA blends (0.5:0, 0.65:0), we find that $A_F(y)$ for the DNA-MT composite exhibits minimal $\dot{\gamma}$ dependence for $\dot{\gamma} > 9\,s^{-1}$. This insensitivity to $\dot{\gamma}$, a hallmark of elasticity, suggests that rigid MTs suppress the viscous dissipation enabled by flexible DNA molecules deforming, stretching and reorienting in response to the strain. The inability of MTs to undergo entropic stretching likely also plays a role in suppressing the non-monotonic rate dependence. Namely, the alignment of DNA may indicate center-of-mass motion aligned with the strain as well as entropic stretching from random coils to more extended strain-aligned conformations. We expect stretching to be stronger with higher entanglement density (i.e., stronger for 0.65:0 than 0.5:0), as it requires that each polymer is pulled in the direction of the strain while also being pulled in the opposite direction by the entangling polymers that are trailing or are further from the strain. This mechanism, consistent with Fig. 2c, should become stronger as $\dot{\gamma}$ increases, until it is faster than the entanglement rate $\upsilon_e$, i.e., the rate for each entanglement segment to 'feel' its tube confinement from surrounding entangling polymers[1,62,78,79].

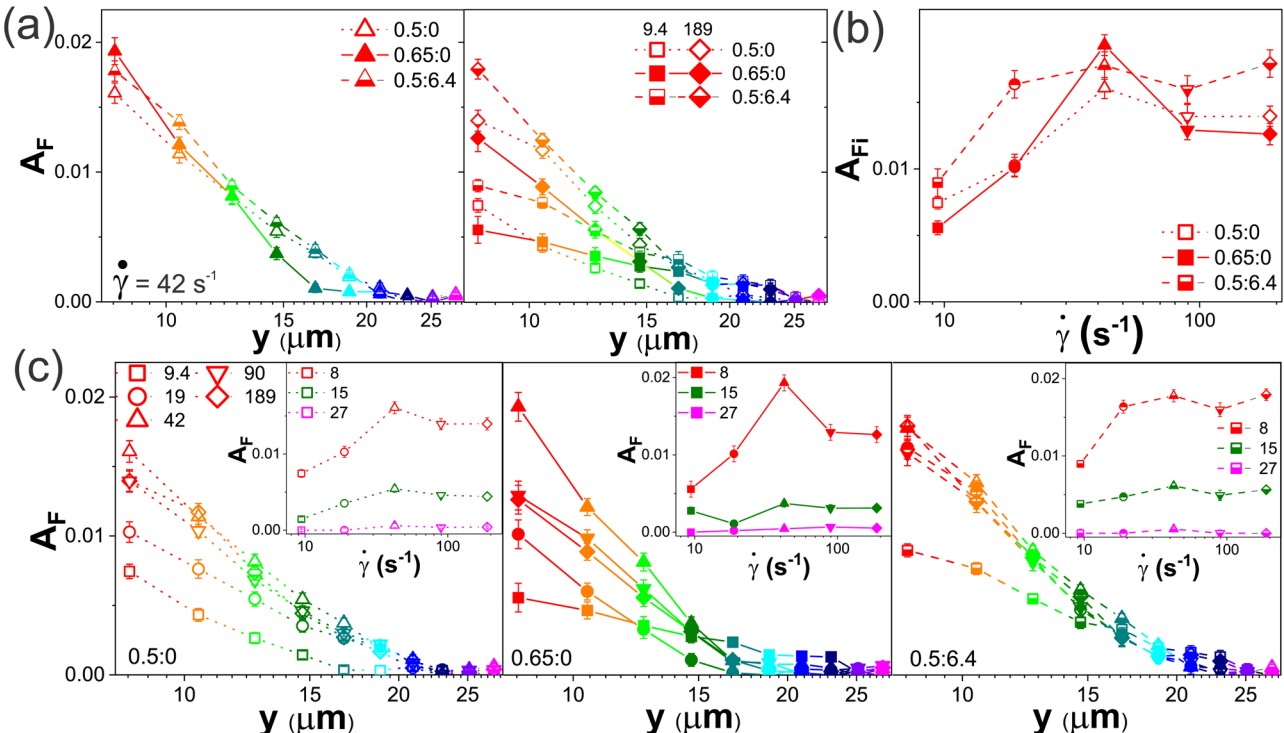

**Fig. 2 | Local shearing induces strain alignment of DNA with a non-monotonic rate dependence. a** Alignment factor $A_F$ versus distance $y$ (increasing from warm to cool colors) for a $\dot{\gamma} = 42\,s^{-1}$ strain (left) and -4.5-fold slower ($\dot{\gamma} = 9.4\,s^{-1}$, squares) and faster ($\dot{\gamma} = 189\,s^{-1}$, diamonds) strains (right) applied to solutions with $c_D : c_T = 0.5{:}0$ (open symbols), 0.65:0 (closed symbols), and 0.5:6.4 (half-filled symbols). **b** $A_{F,i}$, the alignment factor for the closest ROI ($y = 8\,\mu m$), versus $\dot{\gamma}$ for all three systems (listed in legend) reveals a non-monotonic $\dot{\gamma}$ dependence that is most pronounced for 0.65:0 and weakest when MTs are present (0.5:6.4). **c** $A_F(y)$ for

strain rates ($s^{-1}$) of $\dot{\gamma} = 9.4$ (squares), 19 (circles), 42 (upright triangles), 90 (inverted triangles), and 189 (diamonds) for (left) 0.5:0, (middle) 0.65:0, and (right) 0.5:6.4 systems. Insets show the $\dot{\gamma}$ dependence of $A_F$ near ($y = 8\,\mu m$), far ($y = 27\,\mu m$) and at an intermediate distance ($y = 15\,\mu m$) from the strain. Both DNA blends (left, middle) show non-monotonicity that is stronger closer to the strain, while the DNA-MT composite (right) exhibits minimal $\dot{\gamma}$ dependence for $\dot{\gamma} > 9.4\,s^{-1}$. Each data point and corresponding error bar shown in **a–c** represent the mean and standard error across five different videos.

Once the strain surpasses the entanglement rate ($\dot{\gamma} > \upsilon_e$) the polymers no longer have time to feel the tube constraints that facilitate stretching, resulting in less alignment. At the same time, for much lower strain rates ($\dot{\gamma} < \upsilon_e$), the entangled polymers have ample time to relax imposed stress on the timescale of the strain, also resulting in less strain-induced deformation and flow. Only for $\dot{\gamma} \approx \upsilon_e$ are the polymers confined by entanglements but unable to relax imposed stress, leading to pronounced alignment. To quantitatively corroborate this physical picture, we consider the predicted entanglement times $\tau_e$ for the 0.65:0 and 0.5:0 blends: $\tau_{e,0.65} \simeq 15.7\,ms$ and $\tau_{e,0.50} \simeq 26.6\,ms$, respectively (see SI Section S2). These timescales correspond to rates $\upsilon_e = \tau_e^{-1}$ of $\upsilon_{e,0.65} \simeq 64\,s^{-1}$ and $\upsilon_{e,0.50} \simeq 38\,s^{-1}$, which are remarkably similar to $\dot{\gamma} = 42\,s^{-1}$ in which we see maximal strain alignment. Conversely, our faster ($90\,s^{-1}$, $189\,s^{-1}$) and slower ($9.4\,s^{-1}$, $19\,s^{-1}$) rates are $\gtrsim 2\upsilon_e$ and $\lesssim \upsilon_e/2$, respectively.

**DNA exhibits strain-induced superdiffusivity and quiescent subdiffusion at lengthscales dictated by the entanglement density**

We next seek to elucidate the DNA dynamics that give rise to the deformation fields shown in Fig. 2 by evaluating the fluctuation decay times $\tau(q)$ for varying distances $y$ from the strain path as described above (Fig. 1e). We first focus on the $\dot{\gamma} = 42\,s^{-1}$ data for the 0.65:0 blend, as it exhibits the most pronounced strain alignment (Fig. 2). As shown in Fig. 3a, b, $\tau(q)$ curves for all $y$ values collapse to a single power-law curve, $\tau(q) \sim q^{-\beta_2}$, for $q \gtrsim 4\,\mu m^{-1}$, with a weakly subdiffusive scaling exponent of $\beta_2 \simeq 2.6$, suggestive of both weak confinement and minimal strain sensitivity. However, a distinct scaling regime, $\tau(q) \sim q^{-\beta_1}$, emerges for smaller $q$ values with a stark monotonic

decrease in $\beta_1$ as $y$ decreases (cool to warm colors). To better show this effect we scale $\tau$ by $q^{\beta_2}$ (Fig. 3b), such that negative slopes indicate more pronounced subdiffusion ($\beta_2 < \beta_1$) while positive slopes signify superdiffusive or ballistic scaling ($\beta_2 > \beta_1 \rightarrow 1$). As shown, for $y < 15\,\mu m$, $\beta_1$ is indicative of superdiffusion that is more pronounced closer to the strain site, while for $y > 15\,\mu m$, $\beta_1$ becomes larger than $\beta_2$, indicating subdiffusion. This crossover distance ($y \approx 15\,\mu m$) is similar to that at which $A_F$ becomes negligibly small (Fig. 2), suggesting that strain alignment is a result of DNA moving or stretching preferentially (nearly ballistically) in the direction of the strain, but, as $y$ increases, tube confinement begins to dominate the dynamics, resulting in isotropic ($A_F \simeq 0$) and subdiffusive ($\beta_1 > 2$) motion.

To substantiate this interpretation, we recognize that the wavevector $q_c$ at which scaling crosses over from $\beta_1$ to $\beta_2$ corresponds to a lengthscale $\lambda_c = 2\pi/q_c \simeq 1.6\,\mu m$, which is remarkably close to the predicted tube diameter of the 0.65:0 DNA blend, $d_{T,0.65} \simeq 1.5\,\mu m$ (see SI Section S2). At lengthscales smaller than $d_T$, corresponding to $q > q_c$ and $\beta_2$ scaling, DNA segments do not feel the surrounding entanglements so there is both minimal tube confinement (thus weaker subdiffusion), and minimal 'pulling' by entanglements that are moving along with the strain (thus no directed motion).

These general trends in dynamics are also seen for the less entangled DNA blend (0.5:0) and the DNA-MT composite (0.5:6.4; Fig. 2c, d). However, the wavevector at which $\beta_1$ values diverge for different $y$ distances is slightly smaller ($q_c \approx 3.4\,\mu m^{-1}$), as is the corresponding spread in $\beta_1$ values. In other words, the disturbance has less of an effect on these solutions compared to the 0.65:0 blend. The crossover lengthscale in both cases is $\lambda_c = 2\pi/q_c \simeq 1.8\,\mu m$, close the the tube diameter of $d_{T,0.5} \simeq 1.7\,\mu m$ for the 0.5:0 DNA blend (see SI

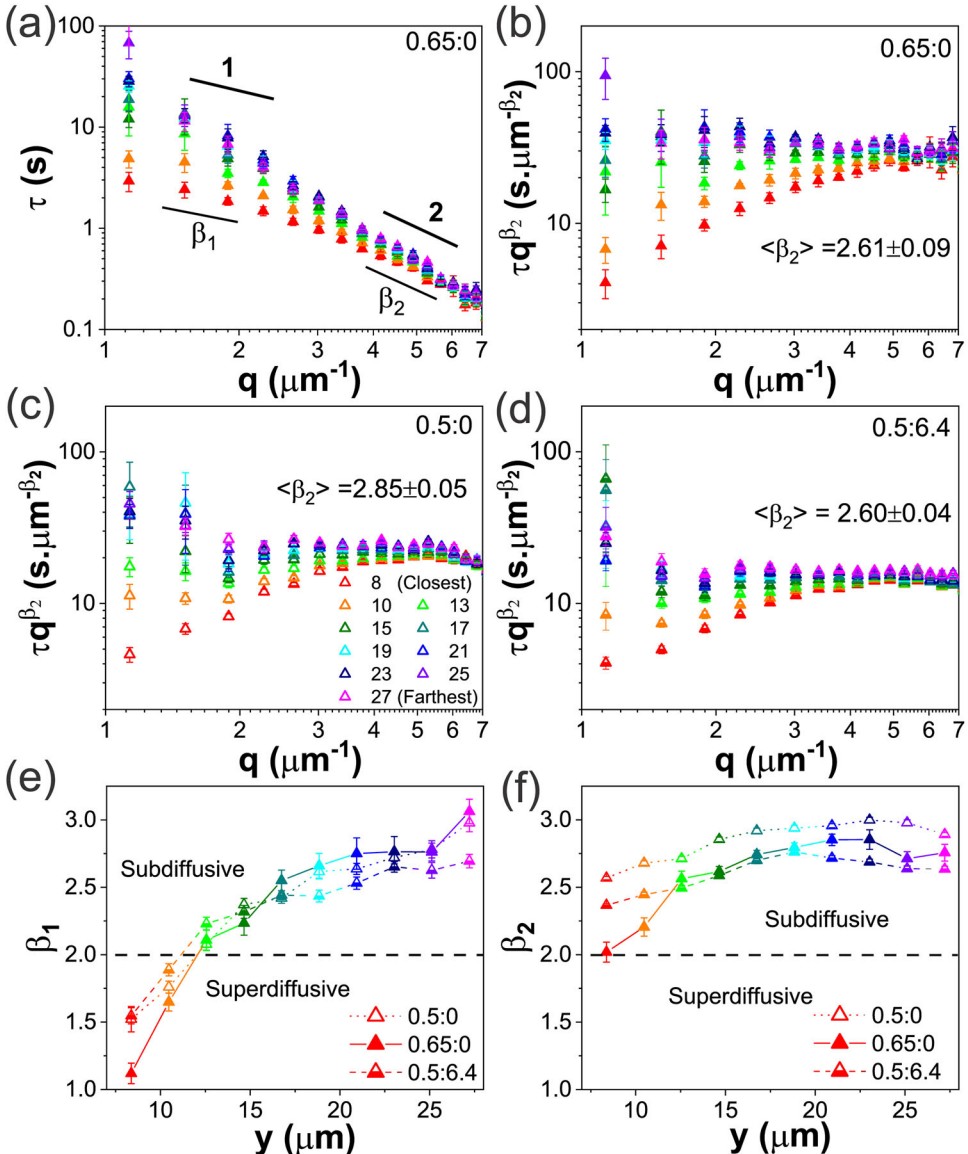

**Fig. 3 | DDM decay times reveal universal biphasic DNA dynamics that range from ballistic to subdiffusive.** All data is for $\dot{\gamma} = 42\,s^{-1}$, which induces the most pronounced alignment (see Fig. 2). **a** DDM decay times $\tau(q)$ versus wavevector $q$ for 0.65:0 evaluated at varying distances $y$, represented as warm to cool colors for $y = 8–27\,\mu m$ (see legend). Closer to the strain (red, orange), $\tau(q)$ curves exhibit two distinct scaling regimes described by $\tau(q) \sim q^{-\beta_1}$ and $\tau(q) \sim q^{-\beta_2}$ for small and large $q$ values, respectively. Scaling exponents for diffusive ($\beta = 2$) and ballistic ($\beta = 1$) dynamics are also shown. $\tau(q)$ curves for $q \gtrsim 4\,\mu m^{-1}$ are largely independent of $y$, while they diverge for $q \lesssim 4\,\mu m^{-1}$. The crossover wavevector $q_c \simeq 4\,\mu m^{-1}$ corresponds to a lengthscale $\lambda_c = 2\pi/q_c \simeq 1.6\,\mu m$, comparable to the tube diameter $d_T \simeq 1.5\,\mu m$ for 0.65:0. **b–d** $\tau(q)$ normalized by the corresponding large-$q$ scaling $q^{-\beta_2}$ versus $q$ for **b** 0.65:0, **c** 0.5:0, and **d** 0.5:6.4 subject to $\dot{\gamma} = 42\,s^{-1}$ straining. All systems exhibit biphasic scaling behavior, and strong $y$ dependence for $q \lesssim q_c$, with $\beta_1$ varying from $<\beta_2$ (positive slopes) to $>\beta_2$ (negative slopes) as $y$ increases. In

contrast, $q \gtrsim q_c$ scaling displays minimal $y$ dependence (data collapse). The crossover from $\beta_1$ to $\beta_2$ scaling occurs at a lower $q_c$ value of ~3.3 $\mu m^{-1}$ for 0.5:0 and 0.5:6.4 compared with 0.65:0, corresponding to $\lambda_c = 2\pi/q_c \approx 1.9\,\mu m$, comparable to $d_T \approx 1.7\,\mu m$ for the 0.5:0 system. **e** $\beta_1$ versus $y$ determined from fitting $\tau(q < q_c)$ curves for all three systems plotted in **b–d**. Dashed line indicates normal Brownian diffusion ($\beta_1 = 2$) while $\beta_1 = 1$ and $\beta_1 > 2$ indicate ballistic and subdiffusive dynamics, respectively. 0.65:0 (closed symbols) exhibits the most superdiffusive, nearly ballistic motion near the strain (small $y$) and most extreme subdiffusion far from the strain (large $y$), whereas 0.5:6.4 (half-filled symbols) exhibits the weakest $y$ dependence and the least deviation from normal diffusion. **f** $\beta_2$ versus $y$, determined from fitting $\tau(q > q_c)$, exhibits minimal $y$ dependence and weakly subdiffusive dynamics. Each data point and corresponding error bar shown in **a–f** represent the mean and standard error across five different videos.

Section S2), corroborating our interpretation of the 0.65:0 crossover. Moreover, the agreement of $\lambda_c$ with $d_{T,0.5}$ for both the 0.5:0 and 0.5:6.4 solutions, implies that the microtubules have little effect on the lengthscale at which confinement effects are felt. To understand this effect, we recall that the mesh size $\xi$ of the microtubules in 0.5:6.4 composites is $\xi \simeq 0.89 c_T^{-1/2} \simeq 1.1\,\mu m$ (here $c_T$ is in mg/ml) and $\xi \simeq d_{T,MT}$. As such, the $q$ value below which microtubule confinement should impact dynamics corresponds to $q_c = 2\pi/\lambda_c \simeq 6\,\mu m^{-1}$, which is close to the largest $q$ value we can accurately measure. As such, any

microtubule confinement effects would be apparent over the entire $q$ range.

To quantify and compare the biphasic dynamic scaling described above for all three systems, we plot their respective $\beta_1$ (Fig. 3e) and $\beta_2$ (Fig. 3f) exponents as functions of $y$. As shown in Fig. 2a–c, $\beta_2$ values, which represent scaling behavior for lengthscales below $\lambda_c$, show minimal $y$ dependence for all systems. Insofar as the $y$ dependence reflects strain-induced dynamics, this result indicates that below the DNA entanglement lengthscale (recall $\lambda_c \approx d_T$) the effects of the strain

are screened by the surrounding entanglements. We also point out that all $\beta_2$ values are subdiffusive with an average value of $\langle\beta_2\rangle \simeq 2.7$. While subdiffusion, at first glance, seems plausible, as it is typically an indicator of strong confinement, such as in entangled or crowded polymer systems[33,40,41,80,81], recall that $\beta_2$ values quantify the dynamics for lengthscales below the tube diameter, where we actually expect confinement to have little impact. Rather, the DNA should be undergoing unconfined normal Brownian diffusion.

This effect suggests that there is another confinement mechanism that persists across the entire $q$ range ($0.9\,\mu m < 2\pi/q < 6.3\,\mu m$), even in the absence of microtubules. As we discussed in the Introduction, threading of ring polymers by their linear counterparts can lead to strong subdiffusion and caging as well as suppressed relaxation of both rings and linear chains in ring-linear blends[44,53,82,83]. Moreover, the effects of threading are most prominent in blends with comparable concentrations of rings and linear chains, as in the systems we study here[44]. While the displacement of a threaded ring parallel to the contour of the threading linear chain is limited to $d_T$ (similar to an entangled polymer) the lateral displacement of the ring is confined to its radius of gyration $R_{G,R}$ (-0.6 μm for the DNA in our systems[84]). As such, confinement from threading impacts dynamics for lengthscales $\lambda > R_{G,R}$, which correspond to $q \lesssim 10\,\mu m^{-1}$, and encompass our entire $q$ range ($1\,\mu m^{-1} \lesssim q \lesssim 7\,\mu m^{-1}$)[85]. As such, threading is likely the dominant mechanism mediating the small-lengthscale ($q > q_c$, $\lambda < \lambda_c$) subdiffusive scaling quantified by $\beta_2$.

Further evidence for the role of threading in the observed dynamics can be seen in the system-dependent alignment and scaling behavior near the strain site (small $y$). Specifically, previous rheological studies of ring-linear blends have shown that threading of rings by linear chains enhances shear-thinning behavior by facilitating their alignment with shear flow[38,50,54,86]. As such, the degree to which rings are threaded in each system likely mediates the extent to which the DNA polymers exhibit strain alignment and superdiffusivity near the strain site. Within this framework, we expect the 0.65:0 blend, with more total rings and linear chains compared to the 0.5:0 blend, and thus more threading events, to exhibit more pronounced alignment and flow than the 0.5:0 system, as we observe in Figs. 2 and 3.

The connection between threading propensity and alignment is less obvious when comparing the 0.65:0 blend to the 0.5:6.4 DNA-MT composite, as previous studies have reported that DNA can become threaded by MTs[22,40]. However, the higher rigidity and longer length of the MTs suppress thermal reconfiguration into threaded conformations and limits the number of free ends available for rings to thread. Additionally, there are 23% fewer rings in the 0.5:6.4 composite compared to 0.65:0, so correspondingly fewer threading events can occur. Reduced threading is further corroborated by less pronounced small-$\lambda$ subdiffusion (smaller $\beta_2$) compared to the DNA blends (Fig. 3f), which we understand to be principally mediated by threading, as described above.

We now turn to understanding the scaling exponent $\beta_1$, which describes the dynamics at larger lengthscales where entanglements need be considered (i.e., $\lambda \gtrsim d_T$). Far from the strain (large $y$), where its effect is negligible, we expect entanglements to lead to more pronounced subdiffusion, compared to $\beta_2$, as they enhance the degree of confinement (beyond threading). Conversely, near the strain (small $y$), where the imposed strain dominates the dynamics, entanglements should increase strain-aligned motion and stretching, as described in the previous section, manifesting as superdiffusive or ballistic dynamics. This trend is exactly what we observe in Fig. 3e, in which $\beta_1$ transitions from -1 (ballistic) near the strain to -3 (subdiffusive) furthest from the strain. However, this $y$ dependence, strongest for 0.65:0, is weakest for the 0.5:6.4 composite, suggesting that microtubules suppress both quiescent subdiffusion as well as strain-induced alignment and flow. To understand this effect, we look to previous studies of actin-microtubule composites which report that rigid confinement from stiff microtubules leads to less deviation from normal

diffusion of tracer particles, compared to confinement by more flexible actin filaments in which pronounced subdiffusion arises from slow rearrangement of the actin entanglements[22,23,40,68].

The takeaway is that at lengthscales smaller than the tube diameter, threading dominates the dynamics for all systems, while at larger lengthscales entanglements dominate. The onset of entanglement dynamics enhances subdiffusion far from the strain and alignment near the strain, with the 0.65:0 blend exhibiting the strongest threading and entanglement effects.

## Deformation dynamics of DNA blends display emergent resonant coupling with strain rate that is suppressed by microtubules

The reduced $\dot{\gamma}$ dependence of the alignment factor (Fig. 2) and $y$-dependence of the dynamical scaling (Fig. 3) for the 0.5:6.4 composite compared to the DNA blends, is indicative of the high rigidity of the MTs compared to DNA. This rigidity hinders the ability of MTs to deform and reorient in response to strain, or fluctuate on the timescale of DNA fluctuations. Because the relaxation rates of MTs are much slower than the applied strain rates and relaxation rates of DNA, we except the 0.5:6.4 system to exhibit minimal $\dot{\gamma}$ dependence of dynamics as compared to DNA blends, similar to the dependence of $A_F$. In other words, this elastic-like (i.e., $\dot{\gamma}$-independent) behavior is a result of MTs being unable to deform in response to the strain, regardless of $\dot{\gamma}$. The dynamics of the DNA confined in the microtubule cages should likewise show minimal $\dot{\gamma}$ dependence, as their dynamics are dictated by the nature of their confinement. In Fig. 4, we verify this interpretation by examining the $\dot{\gamma}$ dependence of the dynamics and strain propagation shown in Fig. 3.

As shown in Fig. 4a, the same two-phase scaling behavior seen in Fig. 3 is preserved for all strain rates, with $\beta_1$ crossing over to $\beta_2$ at $\lambda_c \approx d_T$. Further, similar to Fig. 3f, $\beta_2$ is largely insensitive to $\dot{\gamma}$ or the details of the system and indicative of subdiffusion with $\langle\beta_2\rangle \simeq 2.7$ (Fig. 4b). $y$-dependent $\beta_1$ values, plotted as a function of $\dot{\gamma}$ for each system (Fig. 4c), show minimal $\dot{\gamma}$ dependence for the DNA-MT composite (0.5:6.4) as compared to the DNA blends, confirming our hypothesis above. Further, the average $\beta_1$ values far from the strain ($y>15\,\mu m$) are largest (most subdiffusive) for 0.65:0 ($\langle\beta_1\rangle \simeq 3$) and smallest (most diffusive) for 0.5:6.4 ($\langle\beta_1\rangle \simeq 2.5$). The former is suggestive of confinement from pervasive threading and flexible entanglements while the latter likely arises from rigid caging by the MT scaffold.

Now, considering the dynamics near the strain, if the superdiffusivity we observe in Fig. 3 is indeed due to strain-aligned polymer motion, deformation, and stretching, then the $\dot{\gamma}$ dependence of $\beta_1$ should mirror that of the alignment factor $A_F$ (Fig. 2). Figure 4c definitively shows that $\beta_1$ for both DNA blends (0.65:0, 0.5:0) display similar non-monotonic $\dot{\gamma}$ dependences as $A_F$, with the most pronounced superdiffusive flow emerging at $\dot{\gamma} = 42\,s^{-1}$, and the $\dot{\gamma}$ dependence and deviation from diffusive scaling being strongest for 0.65:0.

This observation–indicating maximal entropic stretching and flow at a strain rate of $\dot{\gamma} = 42\,s^{-1}$ that is 'resonant' with the entanglement rates $v_{e,0.65}$ and $v_{e,0.50}$–is in line with our interpretation in the previous section. Namely, when $\dot{\gamma}$ is substantially faster than $v_e$, the polymers do not feel entanglements that promote flow alignment and stretching, thereby reducing superdiffusivity. At the same time, for $\dot{\gamma} < v_e$ the polymers have time to relax on the timescale of the strain, resulting in less deformation and affine flow (i.e., superdiffusivity) near the strain site.

This pronounced 'resonant' behavior–in which the system response is maximized when the strain rate is comparable to the intrinsic system relaxation rate–is suppressed by microtubules which, due to their rigidity, have slower relaxation timescales. We can estimate a lower bound for the fastest relaxation time for entangled microtubules based on previous studies of actin-microtubule

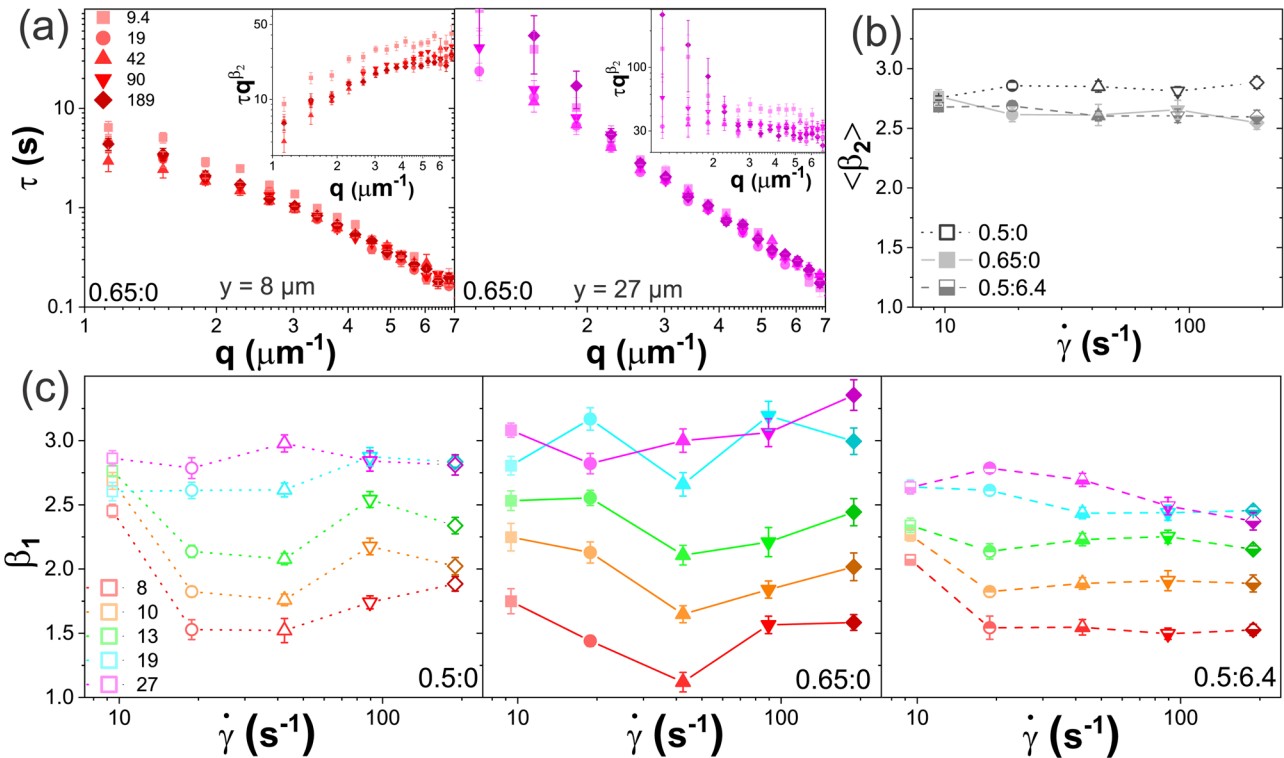

**Fig. 4 | DNA blends exhibit flow-like stress response that is maximized at 'resonant' strain rates. a** $\tau(q)$ versus wavevector $q$ for 0.65:0 evaluated nearest (left, red, $y = 8\,\mu m$) and farthest (right, purple, $y = 27\,\mu m$) from strains with rates of $\dot{\gamma} = 9.4\,s^{-1}$ (squares), $19\,s^{-1}$ (circles), $42\,s^{-1}$ (upright triangles), $90\,s^{-1}$ (inverted triangles), and $189\,s^{-1}$ (diamonds). Color gradients from light to dark indicating increasing $\dot{\gamma}$. Insets: scaled decay times $\tau q^{\beta_2}$ versus $q$ show the biphasic behavior with (left) superdiffusive motion near the strain site versus (right) subdiffusion far from the strain. **b** Mean $\tau(q)$ scaling exponent $\beta_2$ determined from fits to $\tau(q > q_c)$ and averaged over all $y$ values, show minimal dependence on $\dot{\gamma}$ or system. Error bars indicate the standard error across all $y$ values. **c** $\tau(q)$ scaling exponents $\beta_1$ determined from fits to $\tau(q < q_c)$ versus $\dot{\gamma}$ for varying $y$ distances from the strain, listed in $\mu m$ in the legend and color-coded from warm to cool colors for increasing $y$, for (left) 0.5:0, (middle) 0.65:0, and (right) 0.5:6.4. Each data point and corresponding error bar shown in **a**–**c** represent the mean and standard error across five different videos.

composites with similar microtubule concentrations[39,43]. These studies reported a value of $\tau_e \simeq 60\,ms$, corresponding to $\upsilon_e \simeq 17\,s^{-1}$, which is slower than all but our slowest strain rate ($\dot{\gamma} = 9.4\,s^{-1}$). As such, we expect there to be less alignment (smaller $A_F$) and reduced flow ($\beta_1$ closer to 2) in response to the $\dot{\gamma} = 9.4\,s^{-1}$ strain compared to the higher rates, as we indeed see in Figs. 2c and 4c. At this strain rate, the microtubules are able to partially relax and distribute imposed stress on the timescale of the strain.

**Threading of DNA rings in entangled ring-linear blends leads to confinement-dominated deformation dynamics across broad spatiotemporal scales**

To verify and further elucidate our findings and interpretations above, we examine the scaling exponent $\delta(q)$ of the exponential function we use to fit each intermediate scattering function (ISF), as we describe in Methods (see SI Section S1; Fig. 5). For an isotropic system of freely diffusing particles, $\delta(q) = 1$ for all $q$ values[9,71]. However, the ISFs of confined systems are often better described by a stretched exponential function with $\delta < 1$[23,25,68,71,75], while compressed exponentials with $\delta > 1$[72–74] often better fit the ISFs of systems undergoing active transport. Within this framework, we expect to observe $\delta > 1$ near the strain (small $y$) where the DNA is undergoing active deformation, stretching and flow; whereas far from the strain, where quiescent thermal motion should dominate the dynamics, we expect $\delta < 1$. Moreover, this $y$-dependence should become negligibly small as $q$ becomes larger than $q_c$ ($\lambda < \lambda_c$), as at these lengthscales the polymers do not feel the surrounding entangling polymers that enhance confinement far from the strain (lowering $\delta$) and drive strain-induced stretching and deformation near the strain (increasing $\delta$).

These characteristics are clearly demonstrated for all systems in Fig. 5a, in which the stretching exponents measured at the nominal resonant strain ($\dot{\gamma} = 42\,s^{-1}$), are plotted versus $q$ for varying distances $y$ from the strain. For $q < q_c$, $\delta$ decreases from $\delta > 1$ to $\delta < 1$ as $y$ increases (warm to cool colors), with the 0.65:0 system exhibiting the strongest $y$ dependence and reaching the lowest $\delta$ value (-0.6) among the three systems, indicative of the strongest strain propagation and confinement. Likewise, in line with our description above, for $q > q_c$, the dependence on $y$ becomes negligible and $\delta < 1$ for all systems.

We next examine the $\dot{\gamma}$ dependence of $\delta$ to determine the extent to which this metric is sensitive to the resonant behavior we observe for $A_F$ (Fig. 2) and $\beta_1$ (Fig. 4). Figure 5b, which displays $\delta(q)$ curves closest to the strain for each strain rate, along with the $q$-averaged $\delta$ values versus $\dot{\gamma}$ (see insets), indeed show the signature resonant behavior for the two DNA blends, with the $\dot{\gamma} = 42\,s^{-1}$ strain inducing the highest $\delta$ values, consistent with the most directed transport or flow. Conversely, the DNA-MT composite displays minimal $\dot{\gamma}$ dependence and generally higher $\delta$ values, suggestive of reduced strain-coupling and subdiffusion, respectively. These features corroborate the weaker $\dot{\gamma}$ dependence and comparatively smaller values of $\beta_1$ for the DNA-MT composite (Figs. 3e and 4c).

Importantly, we note that while the trends we observe for $\delta$ and $\beta$ are generally consistent with one another and corroborate our interpretations of the data, there are key differences that highlight the need to analyze both metrics to fully capture the dynamics. One important example can be seen by comparing the 0.65:0 data in Fig. 5a with that of Fig. 4c. As shown in the middle panels of each figure, near the strain (small $y$, red data), $\beta_1$ is the most superdiffusive while the corresponding $\delta$ is the smallest (closer to 1) for this system compared to

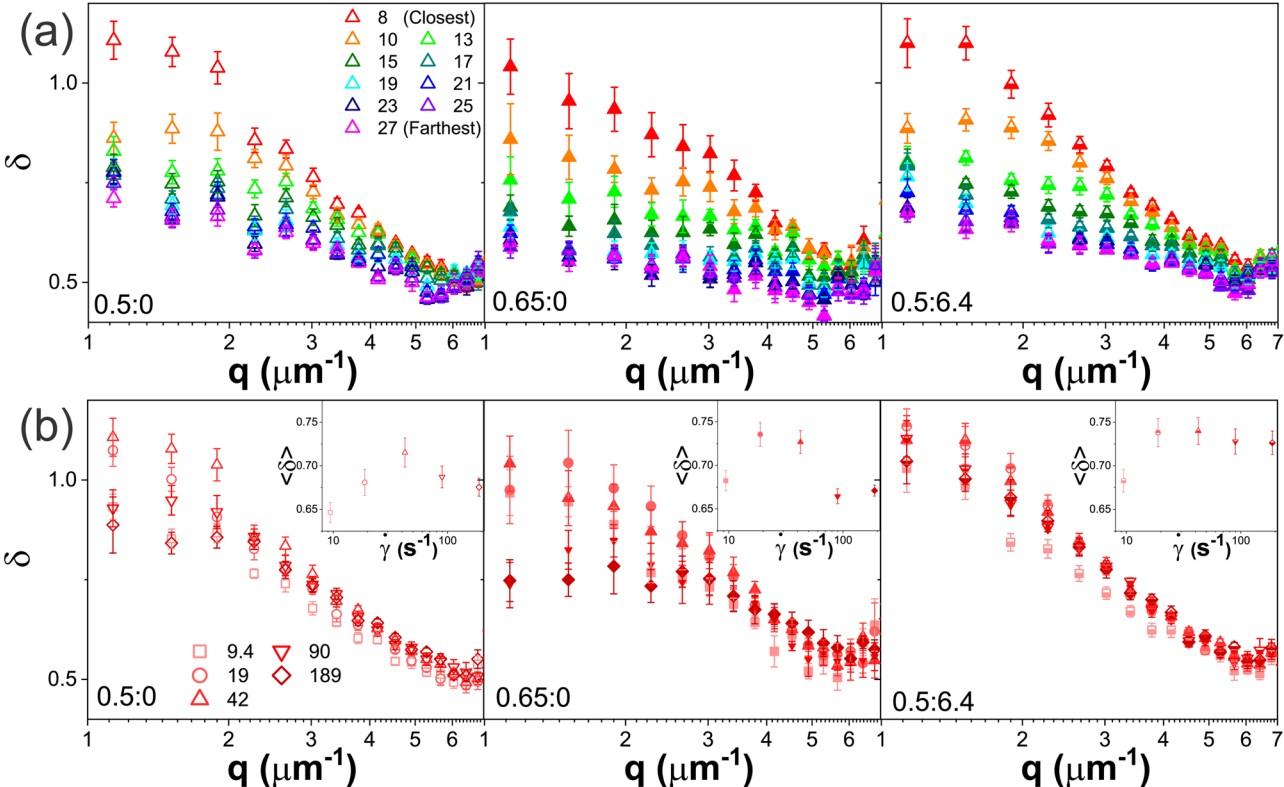

**Fig. 5 | Scaling exponents $\delta$ indicate active DNA transport and confined motion near and far from the strain. a** DDM derived exponential scaling exponents $\delta$ (described in text) versus wavevector $q$ for varying distances from the strain, listed in μm and color-coded from warm to cool colors for increasing $y$, for 0.5:0 (left, open symbols), 0.65:0 (middle, filled symbols) and 0.5:6.4 (right, half-filled symbols) subject to $\dot{\gamma} = 42$ s$^{-1}$ straining. 0.65:0 exhibits the strongest $y$ dependence and lowest values of $\delta$. **b** $\delta(q)$ for $y = 8$ μm (red, closest to the strain) and varying strain rates $\dot{\gamma}$, listed in s$^{-1}$ and delineated by different symbols and a color gradient from light to dark for increasing $\dot{\gamma}$ for 0.5:0 (left, open symbols), 0.65:0 (middle, filled symbols), and 0.5:6.4 (right, half-filled symbols). Insets showing the corresponding $\langle\delta\rangle$, averaged over $q$, versus strain rate highlight the non-monotonic $\dot{\gamma}$ dependence for the DNA-only systems (left, middle). The presence of MTs quenches this non-monotonicity and generally increases $\delta(q)$. Each data point and corresponding error bar shown in **a** and **b** represent the mean and standard error across five different videos.

0.5:0 and 0.5:6.4. While the former observation indicates the most pronounced strain-induced directed transport and deformation, the latter indicates the least degree of active transport (or most pronounced confinement) among the three systems–seeming contradictions. Moreover, far from the strain, $\delta$ values for the 0.65:0 system are lower ($\delta \simeq 0.5$) and exhibit much weaker dependence on $q$ than the other two systems. Taken together, these features suggest that the 0.65:0 system experiences the greatest degree of confinement across all lengthscales (above and below $d_T$) and $y$ distances from the strain, a feature that is not readily obvious in our analysis of $\tau(q)$ and $\beta_1$ and $\beta_2$.

We recognize these differences as indicators of the increased threading probability in 0.65:0 compared to the other systems, in particular the 0.5:6.4 system. As we describe above, threading contributes to the dynamics at all measured lengthscales, whereas entanglements only contribute at $\lambda > d_T$, suggesting that the subdiffusion and confinement that we measure for the 0.65:0 system, in particular, is dominated by threading events. In support of this physical picture, we previously showed that actin-microtubule composites exhibited $\delta$ values of ~0.5 to 0.7 with the lowest and highest values arising in composites that are highly crosslinked and entangled (no crosslinks), respectively[68]. The fact that the quiescent (large $y$) $\delta$ values for the 0.65:0 system are closer to those reported for crosslinked systems, which have much slower relaxation timescales compared to entangled systems, suggests that threaded rings, which also relax much more slowly than entangled rings, play a principal role in the dynamics. This fact, coupled with the comparatively lower $\delta$ values near the strain, indicate that the strain-aligned superdiffusion that we measure is likely due to entropic stretching and deformation of

confined polymers, rather than center-of-mass strain-aligned motion. Notably, this important distinction requires analysis of both $\beta(q)$ and $\delta(q)$.

## Microtubules enhance force response and suppress shear-thinning in DNA blends

The intriguing dynamics described in the previous sections suggest distinct differences in the mechanical response of the 0.5:0, 0.65:0 and 0.5:6.4 systems. For example, the insensitivity of the 0.5:6.4 composite to strain rate is indicative of a more elastic-like response compared to the DNA blends. Further, the resonant dynamics shown in Figs. 2,4,5 may imply a non-monotonic dependence of the stress response on strain rate that would be strongest for the 0.65:0 system and weakest for 0.5:6.4. To explore these questions, we measure the force the polymers exert on the moving probe during the same oscillatory shear program as in Figs. 2–5. Namely, we perform repeated cycles of sweeping the bead back and forth through $s = 15$ μm for a total of 50 s with a $\Delta t_R = 3$ s cessation period between each sweep (Figs. 6 and 7).

We first evaluate the maximum force $F_M$ reached in each sweep for all three systems at four strain rates $\dot{\gamma} = 9.4$–$90$ s$^{-1}$. As shown in Fig. 6a,c, all systems display a monotonic increase in $F_M$ with increasing strain rate $\dot{\gamma}$, in contrast to the non-monotonic $\dot{\gamma}$ dependence of the deformation dynamics (Figs. 2 and 4). Further, the DNA-MT composite displays higher $F_M$ values than DNA-only blends and comparable $\dot{\gamma}$ dependence (Fig. 6a, c), unlike the reduced deformation and minimal $\dot{\gamma}$ dependence of the dynamics (Figs. 2–5). These results suggest that the rigidity and slow relaxation timescales of the MTs, which limit their ability to rearrange and move along with the strain (as

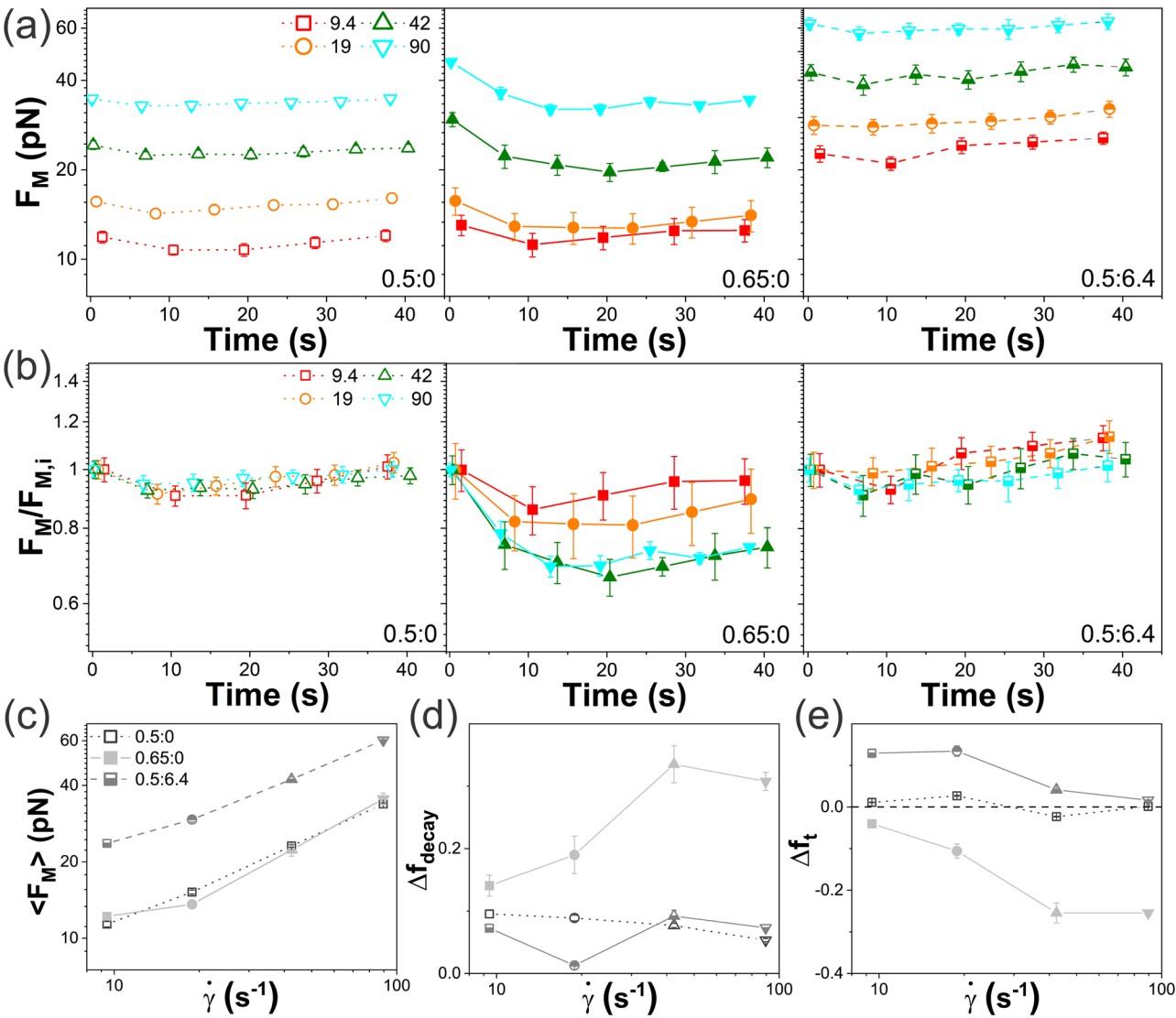

**Fig. 6 | Cyclic force response monotonically increases with $\dot{\gamma}$ and is strongest in the presence of microtubules. a** Maximum force $F_M$ reached during each cycle of repeated nonlinear straining (see Fig. 1) versus measurement time for $\dot{\gamma} = 9.4\,s^{-1}$ (red), $19\,s^{-1}$ (orange), $42\,s^{-1}$ (olive), and $90\,s^{-1}$ (cyan) are shown for 0.5:0 (left, open symbols), 0.65:0 (middle, filled symbols), and 0.5:6.4 (right, half-filled symbols). **b** Data shown in **a**, normalized by the corresponding initial force $F_{M,i}$, shows that $F_M$ for 0.65:0 (middle) decreases with subsequent cycling in a $\dot{\gamma}$-dependent manner while $F_M$ increases for 0.5:6.4 (right). **c** $F_M$ averaged over all cycles, **d** fractional decay in force during cycling, $\Delta f_{decay} = (F_{M,i} - F_{M,min})/F_{M,i}$, and **e** fractional increase in force over the course of cycling, $\Delta f_t = (F_{M,final} - F_{M,i})/F_{M,i}$ versus $\dot{\gamma}$ for 0.5:0 (open), 0.65:0 (filled), and 0.5:6.4 (half-filled). Note that the force generally increases over the course of cycling for 0.5:0 and 0.5:6.4, suggesting polymer buildup at the strain edges, while it decreases for 0.65:0, as expected for shear-thinning and flow alignment. Each data point and corresponding error bar in **a**–**d** represent the mean and standard error across 15 measurements.

evidenced in Figs. 2–4), lead to a strong resistive force response. Conversely, flexible DNA molecules can more easily relax the imposed stress by entropically stretching and configurationally deforming in response to the strain (as shown in Figs. 2–4).

While the initial $F_M$ values for the 0.65:0 DNA blend are higher than 0.5:0 for all strain rates, they subsequently decay to values comparable to those of the 0.5:0 blend. Notably, the fractional drop in force $\Delta f_{decay}$ from the starting value $F_{M,i}$ is greatest for the $\dot{\gamma} = 42\,s^{-1}$ resonant rate (in which polymer deformation and alignment are maximized), with $\Delta f_{decay} = (F_{M,i} - F_{M,min})/F_{M,i} \simeq 0.25$ (Fig. 6b, d). Taken together, these results are indicative of shear-thinning, whereby the viscosity of the solution is reduced in response to strain due to alignment of the polymers with the strain. Indeed, previous studies have shown that entangled equal-mass ring-linear blends exhibit enhanced shear-thinning compared to their pure linear or ring counterparts due to pervasive ring threading[38,76].

The decay in $F_M$ for 0.65:0, which occurs over a time $t_r \simeq 20\,s$, suggests that the longest relaxation timescale of the network has not been reached in the early cycles. Indeed, the predicted disengagement time for linear DNA at 0.65 mg/ml is $\tau_D \simeq 23\,s$ (see SI Section S2), remarkably close to $t_r \simeq 20\,s$ in which $F_M$ reaches a nearly constant time-independent value. This time-independence for times longer than $\tau_D$ suggests that the DNA can relax stress to the same degree, regardless of time, such that a steady-state is reached.

For the less entangled DNA system (0.5:0), $F_M$ values remain nearly constant for all cycles, dropping by <10% in the first 2 cycles, followed by a slight increase that is most pronounced for the slowest strain rate (Fig. 6d), that likewise exhibits the weakest alignment and superdiffusivity (Figs. 2 and 4). To understand the subsequent rise in force following the initial decay, we turn to the DNA-MT composite results shown in Fig. 6. Rather than thinning, which is minimal for 0.5:6.4 (Fig. 6d), we instead observe a modest increase in force over

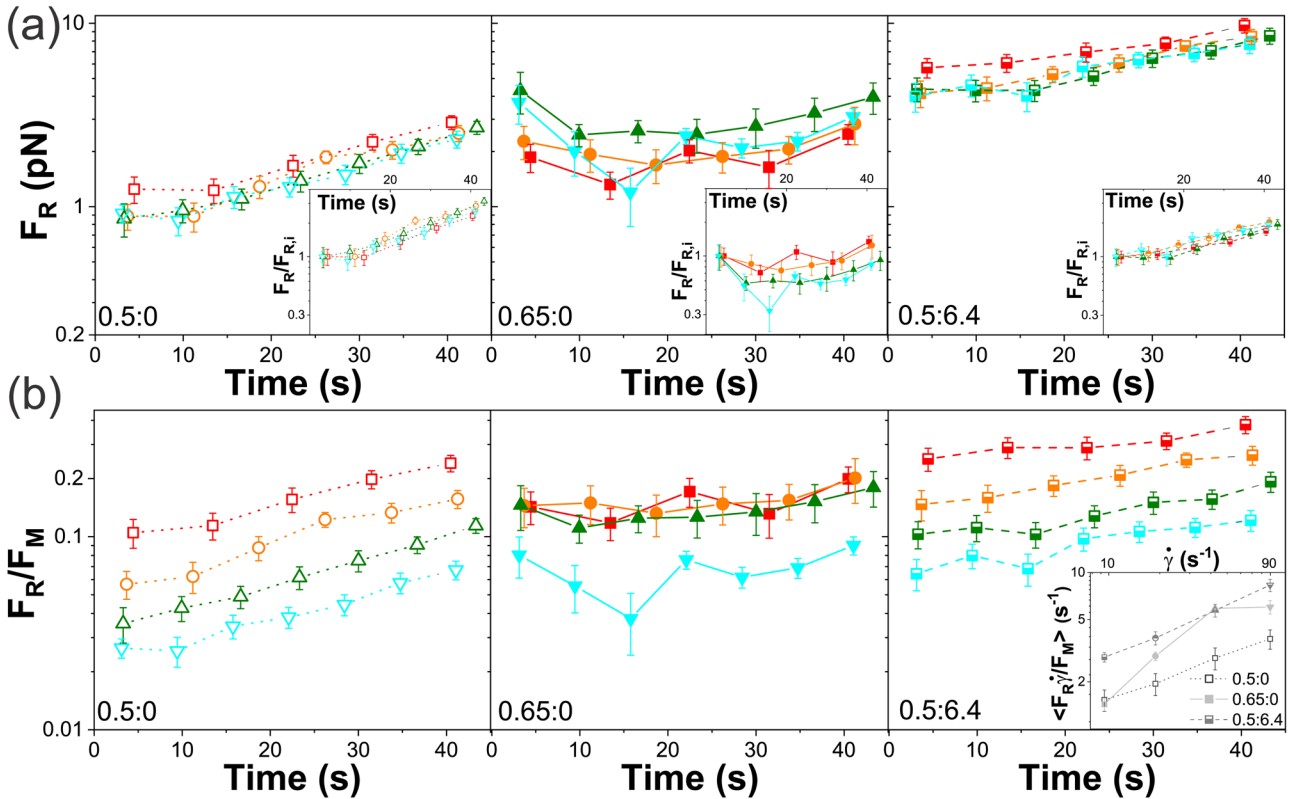

**Fig. 7 | Residual force following cessation is $\dot{\gamma}$-independent with distinct time dependence for the 0.65:0 blend. a** Residual force $F_R$ maintained at the end of each $\Delta t_R = 3$ s cessation period versus measurement time during repeated straining (see Fig. 1) at $\dot{\gamma} = 9.4\,\text{s}^{-1}$ (red), $19\,\text{s}^{-1}$ (orange), $42\,\text{s}^{-1}$ (olive), and $90\,\text{s}^{-1}$ (cyan) for 0.5:0 (left, open symbols), 0.65:0 (middle, filled symbols), and 0.5:6.4 (right, half-filled symbols). Insets show $F_R$ normalized by the corresponding initial value $F_{R,i}$. **b** Data shown in **a**, normalized by the corresponding maximum force $F_M$, shows that the fractional residual force, an indicator of elastic storage, increases with time and $\dot{\gamma}$ for 0.5:0 and 0.5:6.4, while it is nearly constant for the 0.65:0 system, indicating that

different mechanisms are driving the force response and relaxation of 0.65:0 compared to 0.5:0 and 0.5:6.4. The inset shows the time-averaged fractional residual force scaled by the strain rate, $\langle F_R/F_M\rangle\dot{\gamma}$, which is related to the rate of energy dissipation in the system. The larger $\langle F_R/F_M\rangle\dot{\gamma}$ is, the lower the energy dissipation rate, indicating more elastic energy storage. For a system in which an equal amount of energy is dissipated regardless of strain rate, $\langle F_R/F_M\rangle\dot{\gamma}$ is independent of $\dot{\gamma}$. Each data point and corresponding error bar in **a**–**c** represent the mean and standard error across 15 measurements.

the course of cycling (Fig. 6b), reaching $\Delta f_t = (F_{M,final} - F_{M,i})/F_{M,i} \simeq 0.12$ for the slowest strain rate, in which the effect is most pronounced (Fig. 6e). This rate-dependent increase (Fig. 6b, e) is a sign of polymer buildup at the edges of the strain, otherwise described as osmotic compression[64], which has previously been shown in entangled DNA and actin systems[36,64,66]. In other words, DNA polymers cluster at the leading edge of the moving probe, increasing the local entanglement density and forming a wake in the strain path. Threading events, which are most pervasive in the 0.65:0 blend, reduce this clustering by facilitating stretching and alignment which, in turn, reduce polymer buildup at the leading edges[51,86–88]. Reduced threading probability in the 0.5:0 and 0.5:6.4 systems leads to weaker thinning and more polymer buildup, as shown in Fig. 6b, d, e, which is enhanced by rigid microtubules that cage the DNA and suppress network relaxation.

**Interplay between strain-induced stretching and osmotic compression dictates force relaxation and elasticity**

To shed further light on the coupling between the force response, dynamics, and relaxation of the entangled polymers, we evaluate the force relaxation during each $\Delta t_R = 3$ s cessation period between cycles, chosen to be comparable to the longest measurable relaxation time-scale of the systems (see SI Fig. S2). Figure 7a shows the residual force $F_R$ that is sustained at the end of each cessation period as a function of cycling time. Unlike the maximum resistive force $F_M$ reached in each cycle, the residual force $F_R$ is largely independent of $\dot{\gamma}$ for 0.5:0 and 0.5:6.4 and only weakly decreases with increasing $\dot{\gamma}$ for 0.65:0.

Moreover, $F_R$ for the 0.5:0 and 0.5:6.4 systems both increase monotonically with time (i.e., cycle number), whereas the 0.65:0 system exhibits non-monotonic time-dependence with a minimum at $t_r \simeq 20$ s (Fig. 7a), similar to $F_M$ (Fig. 6a, b).

To understand this relaxation behavior, we next evaluate the fractional residual force, $F_R/F_M$ (Fig. 7b), which is a measure of the relative elastic storage of the system. For reference, a completely viscous, dissipative system immediately relaxes all induced stress once the straining stops (i.e., the bead stops moving), in analogy with Stokes drag, such that $F_R/F_M \to 0$ and the polymers store no memory of their initial pre-strained state. Conversely, a purely elastic system maintains the induced stress for as long as the applied strain is sustained (the bead remains fixed at the maximal strain position), maintaining memory of its pre-strained state, such that $F_R/F_M \to 1$.

As shown in Fig. 7b, the 0.5:6.4 and 0.5:0 systems exhibit the highest and lowest $F_R/F_M$ values for all strain rates, indicating that MTs increase elastic storage, likely by suppressing polymer rearrangement, alignment and flow, while the weakly entangled 0.5:0 system can most easily relax induced force via polymer fluctuations. Interestingly, for both of these systems, $F_R/F_M$ increases with increasing time (subsequent cycling) and decreases with increasing $\dot{\gamma}$, which together signify continued buildup and compression of DNA at the edges of the strain path, as described above. In other words, osmotic compression or densification of DNA at the strain edges increases the local entanglement density which, in turn, results in higher residual force and elastic storage (less ability to relax) that continues to increase with

each cycle as more polymers diffuse into the strain wake and get pushed to the leading edge[66].

This process may also explain the $\dot{\gamma}$ dependence of $F_R/F_M$ for 0.5:0 and 0.5:6.4. Namely, slower strains would provide more time for the polymers to rearrange and untangle with neighbors to stay at the leading edge of the probe, thereby increasing $F_R/F_M$; whereas faster strain rates would result in more polymers slipping off of the moving probe as they are pulled back by entangling polymers, thus decreasing $F_R/F_M$.[38,66]

Unlike 0.5:0 and 0.5:6.4, the 0.65:0 system displays minimal dependence on $\dot{\gamma}$ or cycle, suggestive of a different mechanism dictating the local force response; namely, $\dot{\gamma}$-dependent shear thinning and flow alignment facilitated by threading (as described above)[38,76,87]. Specifically, we rationalize that the enhanced threading and preferential alignment with the strain limits the extent to which polymers buildup at the edges of the strain path, thereby negating any dependence on time (cycle number). Likewise, the shear-thinning that manifests from the strain alignment would result in less increase in $F_R/F_M$ as $\dot{\gamma}$ increases, due to the lower local entanglement density arising from polymer stretching and deformation along the strain path.

To provide more evidence for the processes we describe above, and their coupling to the dynamics we observe, we evaluate $\langle F_R/F_M \rangle \dot{\gamma}$ as a function of $\dot{\gamma}$. This quantity is an indicator of the rate of energy dissipation in the system. Specifically, the larger $\langle F_R/F_M \rangle \dot{\gamma}$ is, the lower the rate of energy dissipation, indicating more energy is being stored elastically. For a system in which an equal amount of energy is dissipated regardless of strain rate, $\langle F_R/F_M \rangle \dot{\gamma}$ is independent of $\dot{\gamma}$. However, as shown in Fig. 7c, there is a clear $\dot{\gamma}$ dependence for all systems.

$\langle F_R/F_M \rangle \dot{\gamma}$ increases monotonically with similar scaling for the 0.5:0 and 0.5:6.4 systems, and the magnitude of the DNA-MT composite (0.5:6.4) is ~2–3 fold higher than the corresponding DNA-only system. These trends are in line with the physical picture of polymer buildup at the strain edges that is more pronounced at slower speeds for these systems. As $\dot{\gamma}$ increases, the deformation energy goes increasingly into elastically stretching the polymers rather than the dissipative processes of moving and rearranging the polymers into high- and low-density regions. The higher magnitude for the DNA-MT composite is an indicator of the increased elastic storage that the stiff microtubule network provides, which is independent of $\dot{\gamma}$.

Conversely, the 0.65:0 system displays the non-monotonic 'resonant' dependence that the corresponding DDM data shows (Figs. 2–5). Namely, $\langle F_R/F_M \rangle \dot{\gamma}$ is maximized at $\dot{\gamma} = 42 s^{-1}$, indicating that the rate of energy dissipation is minimized. At this maximum, $\langle F_R/F_M \rangle \dot{\gamma}$ is slightly larger than the DNA-MT composite (albeit within error), whereas at the slowest strain rate, $\langle F_R/F_M \rangle \dot{\gamma}$ for 0.65:0 is slightly lower than for 0.5:0.

These results demonstrate that the processes underlying the force response and dynamics of the 0.65:0 system are distinct from the 0.5 mg/ml DNA systems. In the former case, entropic stretching and flow alignment, with minimal dissipative rearrangement, dictate the dynamics and mechanics; and is most pronounced when the strain rate is resonant with the fastest polymer relaxation rate (i.e., $\dot{\gamma} \approx \upsilon_e$). In the latter case, the data is suggestive of polymer buildup at the leading edges of the probe that arises from reduced network connectivity and strain-coupling. Moreover, these data show that at lower strain rates, the DNA in the 0.65:0 system can move and rearrange to dissipate energy, similar to 0.5:0; whereas at the resonant rate, 0.65:0 deforms more elastically than the network of stiff MTs, in the form of entropic stretching and flow alignment.

## OpTiDDM couples strain-induced stress and viscoelasticity to macromolecular dynamics and alignment in response to local disturbances

To directly couple the macromolecular dynamics we measure using DDM to the local force response we measure using OpT, we evaluate the relationship between key metrics that each measurement

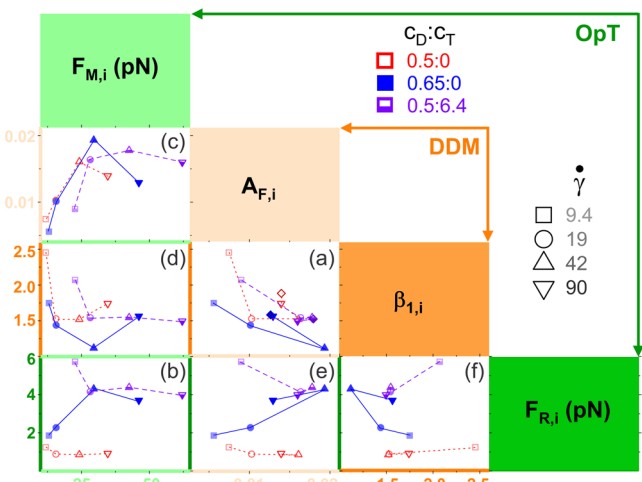

**Fig. 8 | OpTiDDM maps the local stress response to underlying macromolecular dynamics and deformations.** Force-dynamics phase map comparing key metrics from OpT and DDM. $F_{M,i}$ (light green, top left) and $F_{R,i}$ (dark green, bottom right) are the maximum force $F_M$ and residual force $F_R$ for the first strain cycle, measured via OpT. $A_{F,i}$ (light orange, middle left) and $\beta_{1,i}$ (dark orange, middle right) are the alignment factor $A_F$ and $q<q_c$ scaling exponent $\beta_1$ for the closest ROI ($y = 8 \mu m$), determined via DDM. The resulting six unique metric pairs plotted – **a** $(\beta_{1,i}, A_{F,i})$, **b** $(F_{R,i}, F_{M,i})$, **c** $(A_{F,i}, F_{M,i})$, **d** $(\beta_{1,i}, F_{M,i})$, **e** $(F_{R,i}, A_{F,i})$, **f** $(F_{R,i}, \beta_{1,i})$ – are denoted by the axes colors. In all plots the data shown is for 0.5:0 (red, open), 0.65:0 (blue, filled), and 0.5:6.4 (purple, half-filled) subject to strain rates (s$^{-1}$) of $\dot{\gamma} = 9.4$ (squares, light shades), 19 (circles, medium-light), 42 (triangles, medium-dark), and 90 (inverted triangles, dark). Panels **a** and **b** compare metrics from either OpT (**a**) or DDM (**b**). Panels **c–f** each compare an OpT-measured metric ($F_{M,i}$ or $F_{R,i}$) to a DDM-determined metric ($\beta_{1,i}$ or $A_{F,i}$).

produces: the maximum force reached in the initial strain cycle $F_{M,i}$, the residual force at the end of the first cessation period $F_{R,i}$, the alignment factor closest to the strain site $A_{F,i}$, and the small-$q$ ($\lambda > \lambda_c$) scaling exponent $\beta_1$ closest to the strain site ($\beta_{1,i}$). In Fig. 8a–f we directly compare each of these metrics for all strain rates and systems, resulting in six unique plots: $(\beta_{1,i}$ vs $A_{F,i})$, $(F_{R,i}$ vs $F_{M,i})$, $(A_{F,i}$ vs $F_{M,i})$, $(\beta_{1,i}$ vs $F_{M,i})$, $(F_{R,i}$ vs $A_{F,i})$, and $(F_{R,i}$ vs $\beta_{1,i})$. The parameters listed along the top-left to bottom-right diagonal correspond to the x and y axes of each plot, which are color-coded accordingly. We first examine the plots that compare two parameters from the same measurement technique, characterizing either dynamics ($\beta_{1,i}$ vs $A_{F,i}$, Fig. 8a) or force ($F_{R,i}$ vs $F_{M,i}$, Fig. 8b). These plots corroborate what we have already described in the preceding sections, which we summarize as follows.

Figure 8a shows that higher/lower $\beta_{1,i}$ values pair with lower/higher $A_{F,i}$ values. Moreover, while 0.5:6.4 exhibits the smallest spread in $\beta_{1,i}$ and $A_{F,i}$ values for different strain rates, 0.65:0 has the largest spread as well as the highest and lowest values of $A_{F,i}$ and $\beta_{1,i}$, respectively. Higher $A_{F,i}$ values indicate more alignment with the strain, which we understand to arise from increased entropic stretching and motion along the strain direction. As such, increasing $A_{F,i}$ should correspond to $\beta_{1,i}$ values that decrease from >2 (isotropic subdiffusion) to nearly 1 (strain-aligned superdiffusion). Moreover, the increased rigidity and slow relaxation rates of the microtubules in the 0.5:6.4 system reduces the viscous-like dependence of the dynamics on $\dot{\gamma}$ (amplifying elastic-like contributions), and suppresses entropic stretching and coupling to the strain. Conversely, threading, high entanglement density, and having relaxation rates that are comparable to $\dot{\gamma}$ provide the 0.65:0 system with the strongest resonant rate dependence and most pronounced alignment.

Figure 8b shows that, in general, the maximum force $F_{M,i}$ increases with increasing $\dot{\gamma}$, while the residual force $F_{R,i}$ is largely

independent of $\dot{\gamma}$. Moreover, the 0.5:6.4 system exhibits the largest force response (highest $F_{M,i}$) and least stress relaxation (highest $F_{R,i}$), both signatures of enhanced elasticity, likely due to the rigidity of the MTs. Conversely, 0.5:0 displays the lowest $F_{M,i}$ and $F_{R,i}$ values, indicative of its lower entanglement density that allows the polymers to more easily move and rearrange to relax strain-induced force. Moreover, for both 0.5:0 and 0.5:6.4, $F_{M,i}$ increases monotonically with $\dot{\gamma}$ while $F_{R,i}$ is nearly independent of $\dot{\gamma}$. Conversely, the 0.65:0 system, which also shows monotonic increase of $F_{M,i}$ with $\dot{\gamma}$, exhibits a ~2-fold increase in $F_{R,i}$ as $\dot{\gamma}$ increases to the resonant rate $\dot{\gamma} = 42 \text{s}^{-1}$, reaching a value larger than the corresponding value for 0.5:6.4, followed by a modest decrease. Notably, for $\dot{\gamma} < 42 \text{ s}^{-1}$ the residual force for 0.65:0 is comparable to the less entangled DNA system (0.5:0) whereas for $\dot{\gamma} \geq 42 \text{ s}^{-1}$, $F_{R,i}$ for 0.65:0 closely matches that of the DNA-MT composite (0.5:6.4). These data demonstrate a shift in the 0.65:0 data from largely viscous-like (dissipative), similar to the minimally-entangled 0.5:0 system, to more elastic-like, similar to the 0.5:6.4 system, for strain rates comparable to the entanglement rate, $\dot{\gamma} \simeq v_e$, just as the trends in $\langle F_R/F_M \rangle \dot{\gamma}$ show (Fig. 7b).

Having established our chosen metrics as good markers for the dynamics (Fig. 8a) and mechanics (Fig. 8b) we reveal in Figs. 2–7, we next evaluate the plots that compare dynamics to mechanics (Fig. 8c–f). Comparing the alignment factor $A_{F,i}$ to the maximum force $F_{M,i}$ (Fig. 8c) reveals that for slow strain rates, $A_{F,i}$ generally increases with increasing $F_{M,i}$ until $\dot{\gamma} = 42 \text{ s}^{-1}$ followed by a decrease. The non-monotonic dependence of $A_{F,i}$ on $F_{M,i}$ is strongest for 0.65:0, demonstrating maximal strain-aligned deformation and stretching. Also notable is the fact that while $A_{F,i}$ values for 0.5:6.4 are comparable to those for 0.5:0, the corresponding $F_{M,i}$ values are substantially higher, corroborating the fact that microtubules require substantially higher forces than DNA for similar deformation.

We see similar relations between $\beta_{1,i}$ and $F_{M,i}$ (Fig. 8d), with $\beta_{1,i}$ generally decreasing with increasing $F_{M,i}$ until $\dot{\gamma} = 42 \text{s}^{-1}$ after which $\beta_{1,i}$ increases with $F_{M,i}$ for the DNA-only blends (0.5:0 and 0.65:0) while it remains constant for the DNA-MT composite (0.5:6.4). Further, 0.65:0 displays the most extreme non-monotonicity in $\beta_{1,i}$ with increasing $F_{M,i}$ as well as the lowest $\beta_{1,i}$ values, indicating the highest degree of strain-coupling and directed stretching and deformation. This strain alignment, in turn, results in shear-thinning, i.e., reduced increase in $F_{M,i}$ with increasing $\dot{\gamma}$.

How the varying degrees of alignment ($A_{F,i}$) and directed motion ($\beta_{1,i}$) impact the elastic retention of force following the strain ($F_{R,i}$) can be seen in the Fig. 8e, f, respectively. Similar to Fig. 8c, d, $A_{F,i}$ and $\beta_{1,i}$ exhibit mirror dependences on $F_{R,i}$, with all systems displaying non-monotonic $\dot{\gamma}$ dependence, with maxima and minima in $A_{F,i}$ and $\beta_{1,i}$, respectively, occurring at $\dot{\gamma} = 42 \text{s}^{-1}$. Further, the 0.65:0 system displays the broadest range in $A_{F,i}(\dot{\gamma})$ and $\beta_{1,i}(\dot{\gamma})$ values, indicating the largest dynamic range and strongest resonant strain coupling. Moreover, 0.65:0 switches from exhibiting behavior similar to the 0.5:0 blend to that of the 0.5:6 composite. The latter result once again demonstrates a switching from dissipative (small $F_{R,i}$) to elastic-like (large $F_{R,i}$) dynamics for the 0.65:0 blend at $\dot{\gamma} \approx v_e$, while the other two systems display either dissipative (0.5:0) or elastic-like (0.5:6.4) behavior across all $\dot{\gamma}$.

Intriguingly, $A_{F,i}$ decreases (and $\beta_{1,i}$ increases) with increasing $F_{R,i}$ for 0.5:0 and 0.5:6.4 whereas the opposite trend is seen for the 0.65:0 system. We can understand this phenomenon as arising from the varying degree to which DNA builds up or is compressed at the leading edge of the probe versus preferentially aligning with the strain path. Namely, reduced alignment (smaller $A_{F,i}$) coupled to increased elastic storage (larger $F_{R,i}$), as seen for 0.5:0 and 0.5:6.4, indicates that it is polymer buildup at the strain edges, which increases the local entanglement density that, in turn, reduces the ability of the polymers to deform and move in response to the strain (reducing/increasing $A_{F,i}/\beta_{1,i}$). This phenomenon increases the polymer relaxation times,

resulting in reduced relaxation and increased ability to elastically maintain induced force (increasing $F_{R,i}$).

Conversely, the coupled increase of alignment ($A_{F,i}$) and flow ($\beta_{1,i}$) with elasticity ($F_{R,i}$) for 0.65:0 corroborates that enhanced strain-aligned deformation and stretching (increasing/decreasing $A_{F,i}/\beta_{1,i}$) arises from increasingly strong threadings and entanglements which increase the elasticity ($F_{R,i}$). The strong entanglements prevent polymer compression and support entropic stretching, as described above, by polymers being pulled in the direction of the strain from entanglements with leading polymers and opposite the strain from entanglements with trailing polymers. The net result is increased entropic stretching and affine deformation, which, in turn, suppresses buildup at the strain edges and promotes shear-thinning.

## Discussion

Here, we present a robust experimental approach that couples macromolecular dynamics with local force response and strain propagation in polymeric fluids. OpTiDDM (Optical Tweezers integrating Differential Dynamic Microscopy) achieves this coupling by simultaneously imposing local nonlinear strains, measuring the forces the polymers exert to resist the strain, and visualizing the polymers surrounding the strain path. Our DDM analyses map the polymer deformation field as a function of distance from the strain path ($y$), correlation lengthscale ($\lambda = 2\pi/q_c$), strain rate ($\dot{\gamma}$) and network composition ($c_D : c_T$). We quantify the degree of polymer alignment with the strain path ($A_F$) and couple this alignment to the polymer relaxation dynamics (i.e., $\tau(q)$, $\beta$, $\delta$).

We discover a non-monotonic resonant response for entangled DNA blends, in which strain-alignment, superdiffusive motion, and elastic retention of induced force are maximized when $\dot{\gamma}$ is comparable to the entanglement rate $v_e$. Above or below this resonant rate, and for less entangled DNA, these effects are reduced. Incorporating microtubules into the DNA blends suppresses this resonance, as well as the $\dot{\gamma}$ dependence of polymer alignment and superdiffusivity, while at the same time substantially increasing the resistive force and elastic force retention. We rationalize our results as arising from the varying propensity for polymers to build up at the edges versus align and entropically stretch along the strain path, likely dictated by the coupled effects of polymer entanglements, threadings, and the varying relaxation rates and mechanisms accessible to flexible DNA versus rigid microtubules. Importantly, our observations and interpretations of the intriguing physical phenomena that we present here require the direct coupling of macromolecular dynamics and deformation fields to stress response–made possible by coupling OpT microrheology with DDM.

We acknowledge that the systems we use to demonstrate the efficacy and applicability of OpTiDDM are quite complex, such that our interpretations are at times speculative and warrant future studies to test the hypothesized mechanisms. However, a primary goal of this study is to demonstrate how rich of a phase space of physical properties and relationships one can extract from OpTiDDM, and how well one can tease apart and couple together different observed phenomena in even the most complex of systems. Further, wherever possible, we position our discussion of physical mechanisms driving our results in the context of general polymer physics theory and rheology principles, such that our results can be generalized to other polymeric systems and users can draw from our arguments to make sense of their own results on other polymeric systems.

The precise and rich multi-scale characterization and coupling of mechanics and dynamics that OpTiDDM affords–along with its ability to map polymer deformation fields and strain propagation resulting from programmable local disturbances–represents a major advance in the study of soft matter, complex fluids, polymer networks, viscoelastic materials, and even biological cells. With its robust suite of capabilities, as well as its modular and adaptable design, we anticipate

broad interdisciplinary use of OpTiDDM to elucidate non-trivial phenomena that dictate and are impacted by the propagation of localized stresses through a system–critically important to commercial materials applications and cell mechanics alike.

## Methods

We describe many of our methods and approaches throughout the previous sections and in Fig. 1. Expanded description of our methods and materials are provided in Supplementary Information Section S1.

### Reporting summary

Further information on research design is available in the Nature Research Reporting Summary linked to this article.

## Data availability

All data presented in the paper is available via the Science Data Bank repository at https://www.scidb.cn/s/Nru6zi. All raw data will be made freely available upon request.

## Code availability

All custom algorithms used to analyze data can be found on GitHub at https://github.com/kpeddire/OptiDDM.git or made available upon request.

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

## Acknowledgements

This research was funded by grants from the Air Force Office of Scientific Research (AFOSR- FA9550-17-1-0249, AFOSR-FA9550-21-1-0361) awarded to R.M.R.A.

## Author contributions

R.M.R.A. conceived the project, guided the experiments, interpreted the data, and wrote the manuscript. K.R.P. designed and performed the experiments, analyzed and interpreted the data, and wrote the manuscript. R.C. and P.N. helped prepare samples, perform experiments, and analyze data. R.J.M. developed analysis software and helped analyze data and write the manuscript.

## Competing interests

The authors declare no competing interests.
