## [Peer Review File · Nature Communications]

OpTiDDM maps the spatiotemporal propagation of nonlinear strains in polymer blends and compositesREVIEWER COMMENTS

Reviewer #1 (Remarks to the Author):

In this article the authors present an experimental method that combines two well established techniques often used for performing microrheology, i.e. Optical Tweezers and Differential Dynamic Microscopy.

Although I appreciate the authors' ingenious in combining these techniques for microrheology purposes, I have my doubts that the current manuscript satisfy the requirements for publication in Nature Communication, as none of the (debatable) results presented in this work represents a strong case to justify the effort in putting together such an instrumentation.

Therefore, I would suggest to transfer the manuscript to another journal of the Nature family, such as Scientific Reports, for which novelty is not a requirement.

However, before reconsideration, the authors should address the following point of criticisms:

1) It would be appreciate if they could clarify the differences between the proposed method "OpTiDDM" and "OpTIMuM" recently published by Matheson et al. Scientific Reports 11, 5614 (2021). <https://doi.org/10.1038/s41598-021-85013-y>

2) The manuscript is very confusing, too much information with little background and too wordy, not a good example of scientific writing.

3) The figures are clearly very colourful, but their content is not informative; the worst is Figure 8.

Reviewer #2 (Remarks to the Author):

Review of OpTiDDM

The authors combine optical tweezer force measurement with DDM analysis of the surrounding polymer motion to present a very important technique and as an example of its power they report novel physical results in a complex mixed polymer system. Overall I think this work is exciting and paves the way for powerful applications of optical tweezers and DDM. The manuscript is well-written and has far reaching implications, suitable for Nature Communications.

Specific points for NCOMM:

- What are the noteworthy results? - A combination of two powerful techniques (OT and DDM) used to make novel mechanical observations in complex mixed polymer systems.
- Will the work be of significance to the field and related fields? How does it compare to the established literature? If the work is not original, please provide relevant references. - This work will impact the field of polymer rheology as well as several other fields that use OT and DDM. The work is original and a significant advance from previous studies.
- Does the work support the conclusions and claims, or is additional evidence needed? - The work does support the claims without further experiments being necessary.
- Are there any flaws in the data analysis, interpretation and conclusions? Do these prohibit publication or require revision? There are no flaws in the data analysis, interpretation, or conclusions. However the manuscript clarify may be improved if the authors wish to consider my comments below.
- Is the methodology sound? Does the work meet the expected standards in your field? - Yes
- Is there enough detail provided in the methods for the work to be reproduced? - Yes

Below I include comments and questions that I hope that authors will consider addressing, which may help improve the understandability of this work to the wide audience of Nature Communications.

Comments and Questions:

- pg1: In the title, abstract, and throughout the manuscript two phrases are used that could use some clarification: (1) "map stress propagation" - If I understand correctly, stress propagation (in terms of how the stress field evolves in space and time) is not actually mapped in this work, but rather the displacement field is characterized in space due to a localized force application (and measurement). To get from a deformation field to a stress field would require many assumptions (e.g. strain

compatibility being fulfilled and a constitutive relation between stress-strain). If this is true, I think the wording should be modified throughout to clarify this. (2) "elastic memory" - I am not sure what this exactly means since in simple terms an elastic material should not have any memory? And if it does have memory (described by some kernel) then it would then be a viscoelastic material? Some clarification of what the authors mean by this term would be helpful.

- pg4: when moving the sample (not trap), can you also move your DDM observation window the same as the stage motion so you would be in the frame of reference of the sample? Is there any value to doing this?

- pg4: How does the sample thickness affect these measurements and analysis? It seems like thicker samples would attenuate the DDM signal because there would be averaging over the sample depth. Is that true?

- pg4: For the waiting time, $\Delta t_R = 3s$, how was this chosen does the system continue to relax after this time? How would users go about choosing this time for a different sample?

- pg4: How is the density of labeling chosen? How does one determine a good signal?

- Fig 1c: why did you choose this specific Δt for the D matrix? Should users be looking for something specific?

- pg6: How are beta and delta related? They seem like two independent parameters describing the same types of behaviors in f or τ . For instance, could one have $\beta = 1$, ballistic, but $\Delta = 0.5$, super diffusive in the same fit? I realize these parameters were not introduced for the first time in this paper, so details aren't necessary, but some discussion would be helpful.

- Fig2: The A_F is a very small number, can the authors give some feeling for what that means in terms of alignment? Is that like % aligned vs random?

- pg 8-10 and fig 3: It is a bit confusing to call affine motion near applied deformation as super diffusive or subdiffusive. Is there any way to relate this language or make an analogy to the usual description continuum description of the expected deformation field in an elastic/viscous continua like $1/r^n$. The more I think about this the more confusing it gets, because the term diffusive suggests random motion, which is almost the opposite of affine. Can the authors comment on this and explain this disconnect between random diffusion and affine deformation? Presumably this has something to do with the coarse graining inherent in DDM.

- p11: β_1 is like some metric for strain it seems because it is sensitive to larger length scales, whereas β_2 is for smaller length scales where only microscopic non affine motion is occurring? Is this correct? Any comment is welcome.

- p11: I am struggling to somehow relate the betas to strain. I am not sure if there is a good relationship here or if the authors can explain. At some point in the manuscript, it would be helpful to explicitly point out what the power of DDM is over simply doing PIV to get a strain field. This may help clarify some confusion and highlight the authors results.

- p11: The details are good, but almost overwhelming, a contextual paragraph before and after would be very welcome

- p15-16: The monotonic force behavior and non-monotonic strain behavior is almost begging for a physical model. Do the authors know of any possible material models to explain this?

- p16: some of the discussion here about polymer buildup or leaving wakes behind seem like they could be analyzed using the authors technique, by applying DDM at these locations. If this is true, perhaps the authors can comment on how this may be done in the future if they dont want to include it

in this study.

- p16: Force dissipation is a bit of a confusing term. Assuming the force dissipates over time, then there is a timescale (or velocity) involved and then it is usually discussed as an energy dissipation ($F \cdot v$). Elastic memory is also a confusing term - seems like elastic should be viscoelastic? I don't understand how FR/FM is a measure of relative elastic memory versus dissipation. It just seems like a measure of dissipation alone, which would also depend on the strain rate because $P = F \cdot v$. The lower the FR/FM, the lower the residual force is - meaning the most "force" was dissipated - so it just looks like higher strain rate leads to more "force" dissipation? — which maybe leads to an equal amount of energy dissipated in $F \cdot v$? Any clarification or comment here is welcome.

- pg 18: The statement about what slower and fast strain rates lead to could use a reference if this is known in the literature. If not, and it is conjecture, this should also be made clear.

- pg 21: is there a way to generalize the observations in these DNA blends to general polymer systems? And perhaps clear implications of these findings?

- pg 23: are DDM measurements and force measurements done simultaneously?

- pg 23: Explain how $\Delta(t) = 3s$ was chosen.

- pg 24: Can any interpretation for the value of $A_F = 0.02$ with radial averaging be provided? The number is very low (maybe lower due to radial averaging), so can the authors say a bit more about what they think this means?

- pg 24, If I am interpreting the parameters in the ISF correctly then you end up with an $\exp(-\Delta(t) / \tau(-\beta * \Delta))$, so then there are two competing parameters (β and Δ) that are multiplied here. But it seems possible to determine them independently, can the authors comment on this point?

Reviewer #3 (Remarks to the Author):

The authors perform experiments in which they combine two experimental tools that they had previously employed (Optical Tweezers and Differential Dynamic Microscopy) to map the response to local nonlinear strains in polymeric fluids, namely entangled DNA solutions with and without incorporated microtubules.

Overall, I find that the manuscript has good potential and could eventually be published in Nature communications. The methodology presented is undoubtedly innovative and intriguing, and some of the results obtained appear interesting and promising: among other things, the main result seems to be a synergic response of the DNA blends to perturbations with strain rates of the order of the intrinsic entanglement rate, an effect that seems to be suppressed by incorporation of microtubules.

The present version of the manuscript let the reader clearly understand that the authors did a great amount of work, which seems to be conducted in an appropriate way. Unfortunately, the authors fail to offer a clear description of the main mechanisms behind all their observations, not being true to their promise in the abstract to "unambiguously connect polymer dynamics to force response".

The main problem is that the authors mostly list a large number of experimental findings without providing a clear pathway to their interpretation and leaving many questions unanswered.

In addition, I find the current version very little appealing, I find particularly disappointing the lack of accent in the storytelling, which carries the reader for 24 pages without highlighting in a proper way the key results and the crucial steps in getting such results.

Maybe, the authors should consider moving some of the discussion out of the main text, by keeping

only those parts that constitute the core message of the paper.

Having read the manuscript three times, I am still unable to provide a concise list of changes because the problem with the present manuscript cannot be solved locally but requires a global rewriting that make it fit a multidisciplinary journal, publishing in which requires an effort to present facts and hypotheses in the most possible clear way, by clearly outlining the noteworthy results, and why are they significant in the authors' opinion. Both goals are not reached by the current version.

NCOMMS-22-09238R Response to Reviewer Comments

Reviewer comments are reproduced in *green font*

Excerpts from the revised manuscript are in *blue font*. Not all references are not included in the excerpts (replaced with [...] in some places) but can be found in the manuscript.

Reviewer #1 (Remarks to the Author):

In this article the authors present an experimental method that combines two well established techniques often used for performing microrheology, i.e. Optical Tweezers and Differential Dynamic Microscopy.

Although I appreciate the authors' ingenious in combining these techniques for microrheology purposes, I have my doubts that the current manuscript satisfy the requirements for publication in Nature Communication, as none of the (debatable) results presented in this work represents a strong case to justify the effort in putting together such an instrumentation. Therefore, I would suggest to transfer the manuscript to another journal of the Nature family, such as Scientific Reports, for which novelty is not a requirement.

We thank the reviewer for acknowledging the 'ingenious' of our approach, which is indeed essential to the results we present in this work. We realize that perhaps we did not emphasize enough the significance of the combined approach and the information that it can uniquely provide. In the revised manuscript we make these points more clear.

However, before reconsideration, the authors should address the following point of criticisms:

1) It would be appreciate if they could clarify the differences between the proposed method "OpTiDDM" and "OpTIMuM" recently published by Matheson et al. Scientific Reports 11, 5614 (2021). <https://doi.org/10.1038/s41598-021-85013-y>

We thank the reviewer for drawing our attention to this relevant work which we now discuss in the introduction of the manuscript. While the acronym is similar to ours, the techniques are entirely different. The novelty and power of our approach lies in the combination of imaging and extracting information about the surrounding medium with the measurement of local force. OpTIMuM incorporates multiplane microscopy into an optical tweezers setup to allow for video tracking of a trapped bead in 3 dimensions. This advance is to improve the accuracy of the force detection to discern microrheological properties of the surrounding medium. There is no movement of the bead or straining of the material, nor is there imaging of macromolecules or tracers in the surrounding material. The power of our approach is that we can directly connect the macromolecular dynamics and deformations with the measured force the system exerts in response to straining that we can precisely sculpt. We have added the following statement to the Introduction to acknowledge this work:

'Recent advances to OpT microrheology include, for example, coupling OpT to microfluidics to enable force measurements during in-situ modulation of environmental conditions [...]; and integrating multiplane microscopy into OpT to allow for video tracking of a trapped bead in three dimensions to improve force detection accuracy and discernment of microrheological properties [...].'

2) The manuscript is very confusing, too much information with little background and too wordy, not a good example of scientific writing.

We thank the reviewer for suggesting adding more background and being less verbose. While Reviewer #2 found the manuscript 'well-written', we have nevertheless substantially re-written much of the

manuscript to improve clarity, contextualization, and comprehension. We have removed descriptions of less significant results, as well as redundancies. We have also provided more context and explanation of results. These changes are in blue font throughout the text.

3) The figures are clearly very colourful, but their content is not informative; the worst is Figure 8.

We are unsure what is not informative about the figures and why Figure 8 is 'the worst'. There is an enormous amount of data and information in each of the figures and we worked very hard to present them in the clearest and most digestible manner. The color palette was chosen to be most informative to the reader and allow for facile comparisons between data in different figures. For example we use a rainbow scale to denote increasing distances from the strain path for all figures. We also use consistent symbol shapes and symbol fills to indicate different strain rates and systems for all images. We will happily make specific suggested edits if provided.

Reviewer #2 (Remarks to the Author):

The authors combine optical tweezer force measurement with DDM analysis of the surrounding polymer motion to present a very important technique and as an example of its power they report novel physical results in a complex mixed polymer system. Overall I think this work is exciting and paves the way for powerful applications of optical tweezers and DDM. The manuscript is well-written and has far reaching implications, suitable for Nature Communications.

We thank the reviewer for finding our work 'exciting' and paving 'the way for powerful applications of OpT and DDM'. We are also pleased that the reviewer finds our manuscript 'well-written' and 'suitable for Nature Communications' with 'far reaching implications'. Below we thoroughly address all of the Reviewer's questions and useful suggestions.

Specific points for NCOMM:

- What are the noteworthy results? - A combination of two powerful techniques (OT and DDM) used to make novel mechanical observations in complex mixed polymer systems.

- Will the work be of significance to the field and related fields? How does it compare to the established literature? If the work is not original, please provide relevant references. - This work will impact the field of polymer rheology as well as several other fields that use OT and DDM. The work is original and a significant advance from previous studies.

- Does the work support the conclusions and claims, or is additional evidence needed? - The work does support the claims without further experiments being necessary.

- Are there any flaws in the data analysis, interpretation and conclusions? Do these prohibit publication or require revision? There are no flaws in the data analysis, interpretation, or conclusions. However the manuscript clarify may be improved if the authors wish to consider my comments below.

- Is the methodology sound? Does the work meet the expected standards in your field? - Yes

- Is there enough detail provided in the methods for the work to be reproduced? - Yes

Below I include comments and questions that I hope that authors will consider addressing, which may help improve the understandability of this work to the wide audience of Nature Communications.

Comments and Questions:

1. - pg1: In the title, abstract, and throughout the manuscript two phrases are used that could use some clarification: (1) "map stress propagation" - If I understand correctly, stress propagation (in terms of how the stress field evolves in space and time) is not actually mapped in this work, but rather the displacement field is characterized in space due to a localized force application (and measurement). To get from a deformation field to a stress field would require many assumptions (e.g. strain compatibility being fulfilled and a constitutive relation between stress-strain). If this is true, I think the wording should be modified throughout to clarify this. (2) "elastic memory" - I am not sure what this exactly means since in simple terms an elastic material should not have any memory? And if it does have memory (described by some kernel) then it would then be a viscoelastic material? Some clarification of what the authors mean by this term would be helpful.

We thank the Reviewer for the suggestion to improve the clarity of our terminology.

(1) We agree with the Reviewer that perhaps stress propagation is not entirely accurate. We have changed the language throughout the text to 'deformation field' and 'strain field'. However, we note that we do not actually map a displacement field in the sense that we do not measure the displacement of the tracers. Rather we focus on the dynamics—how the motion of the polymers aligns with the strain and the

characteristics of said motion (i.e., diffusive, subdiffusive, etc). We map these dynamical properties as a function of distance from the strain. Given the complex relationships between polymer dynamics, relaxation, and stress response, we do not believe that a displacement field would provide sufficient insight into the stress-strain relationships.

(2) What we mean by elastic memory is that the stress imposed by the strain does not relax, i.e., energy is not dissipated. The material 'remembers' its initial 'pre-strain' state and can return to it because the energy is stored. We have used this terminology in previous works to describe the extent to which systems are able to relax induced stress, thereby erasing any memory of the previous state; or, conversely, retain the stress, and thus have 'memory' of their previous pre-strain states. However, to avoid confusion we have replaced many references to elastic memory with elasticity or elastic storage. We have also added the following clarifying statement to the manuscript:

'For reference, a completely viscous, dissipative system immediately relaxes all induced stress once the straining stops (i.e., the bead stops moving), in analogy with Stokes drag, such that $F_R/F_M \rightarrow 0$. The polymers store no memory of their initial pre-strained state. Conversely, a purely elastic system maintains the induced stress (storing as potential energy) for as long as the applied strain is sustained (i.e., the bead remains fixed at the maximal strain position), thereby retaining memory of its pre-strained state, such that $F_R/F_M \rightarrow 1$.'

2. - pg4: when moving the sample (not trap), can you also move your DDM observation window the same as the stage motion so you would be in the frame of reference of the sample? Is there any value to doing this?

We thank the Reviewer for the question. From a post-acquisition processing standpoint it is straightforward to move the observation window with the sample. In fact, doing so would be required if we were to perform DDM on videos acquired during our force measurements, in which we move the sample while keeping the trap fixed. If we did not remain in the reference frame of the sample then our DDM analysis would be dominated by the stage motion rather than the motion of the DNA in the sample. However, we do not perform DDM analysis on videos acquired during our force measurements (fixed trap / moving stage) because the mismatch between our video acquisition rate and strain rate (the rate we move the stage) leads to blurred or smeared images. Specifically, the fastest rate at which we can acquire videos with sufficient signal-to-noise is 60 fps, and the speeds at which we move the stage to induce substantial stress and deformation to the sample are $v=10-200 \mu\text{m/s}$. With our camera resolution of 130 nm/pixel, during every frame (1/60th s) the sample moves $v / 0.13 \mu\text{m/px} / 60 \text{ fps} = 0.13 v = 1.3 - 25$ pixels. Because the sample moves substantially more than the pixel size, the images are smeared out, preventing us from being able to accurately perform DDM analysis. For this reason, we move the trap (via a piezoelectric mirror) while keeping the sample (stage) fixed for DDM measurements, so the images we acquire are in the reference frame of the sample. We do not measure force in these measurements as our current force detection optics convolute trap motion and force-induced laser deflection. In the next generation of OpTiDDM experiments we will work towards simultaneous force and DDM measurements. We have revised the beginning section of the results section where we describe our approach to better clarify the specifics of our measurements and the rationale, as follows:

'In our setup, an optically trapped microsphere probe (Fig 1a) can be translated through a sample at precise distances (up to 50 μm) and rates (up to 100's of $\mu\text{m/s}$) using a piezoelectric mirror to move the trap or a piezoelectric stage to move the sample relative to the trap¹⁷. Simultaneously, we can measure the force the sample exerts on the probe by using a position-sensing detector (PSD) to measure the laser deflection, and image the surrounding polymers using the fluorescence microscopy capability. These

features allow us to impart nonlinear and mesoscale strains, measure the resulting local stress response (Fig 1b), and perform DDM analysis to extract macromolecular dynamics and deformations (Fig 1c-g).

For force measurements, we impart strains by moving the stage (keeping the trap fixed) to ensure that the laser deflection is entirely from the force imparted on the probe and not from the moving trap (Fig 1b)¹⁷. This approach is based on our previously established OpT microrheology protocols for measuring linear and nonlinear stress response of a range of polymer networks and soft materials^{11,36,38,42,43,61-66}.

To determine the impact of cyclic straining on the polymer dynamics and map the strain-induced deformation field, we impart strains by moving the trap (keeping the sample fixed) to allow for precise imaging of the thermal motion and strain-induced deformations of polymers in the field-of-view (FOV). Specifically, we use the fluorescence capability of our OpT-enabled microscope to image fluorescent-labeled DNA tracer molecules embedded in the sample and filling a $78 \mu\text{m} \times 117 \mu\text{m}$ FOV centered on the strain path of the probe (Fig 1c), and record time-series of the moving DNA tracers throughout the duration of the straining. We then perform differential dynamic microscopy (DDM) on spatially-resolved regions of interest (ROI), each $(16.6 \mu\text{m})^2$, centered horizontally (along x) with the strain path and vertically (orthogonal to the strain) at distances of $y = 8 \mu\text{m} \approx s/2$ to $y = 27 \mu\text{m} \approx 2s$ from the strain path (Fig 1c).

We use DDM instead of more conventional single-particle-tracking (SPT) and particle image velocimetry (PIV) methods to allow for a high density of labeled molecules that are smaller, dimmer, and more susceptible to photobleaching as compared to microspheres and other standard probes. The high density, which prevents tracking single particles (as is done in SPT and PIV), allows us to spatially resolve statistically robust dynamics within each ROI. Further, a lower signal threshold than is needed for SPT and PIV facilitates using DNA or other macromolecules.'

3. - pg4: How does the sample thickness affect these measurements and analysis? It seems like thicker samples would attenuate the DDM signal because there would be averaging over the sample depth. Is that true?

We thank the Reviewer for raising this question. The Reviewer is correct that, because we use wide-field fluorescence microscopy we do have signal from out-of-focus planes that contribute to our images. We have optimized the density of DNA labels such that they are dense enough for ample statistics in small ROIs while dilute enough such that the signal is not overwhelmed by signal from out-of-focus planes. Nevertheless, to improve signal-to-noise one could use confocal or light-sheet microscopy to eliminate out-of-focus light. There are examples of optical tweezers coupled to these microscopy modalities in the literature (e.g., Refs 93-96) that one could follow to build the necessary instrumentation. We have added the following information to the Methods section to clarify:

'We note that our use of wide-field fluorescence microscopy for imaging has the limitation that signal from out-of-focus planes can contribute to the images. We have optimized the density of DNA labels such that they are dense enough for ample statistics in the small ROIs necessary to achieve sufficient spatial resolution, while dilute enough such that the imaging is not overwhelmed by signal from out-of-focus planes. Nevertheless, should one require increased signal-to-noise, OpT could be coupled with confocal or light-sheet microscopy that eliminate out-of-focus light¹⁰¹⁻¹⁰⁴.'

4. pg4: For the waiting time, $\Delta t_R = 3s$, how was this chosen does the system continue to relax after this time? How would users go about choosing this time for a different sample?

We thank the Reviewer for asking this insightful question. We chose this cessation time to be comparable to the longest measurable relaxation time of the systems we investigate. Below this timescale the

systems would still be relaxing at the onset of the next strain in the cycle, complicating the resulting dynamics. Using a waiting time longer than this timescale would unnecessarily prolong measurements, thereby reducing the number of cycles that we can perform before the signal photobleaches.

To determine this longest measurable relaxation time, we performed single strain experiments (no cycling), similar to those performed in Refs 36,38,39,45, in which we move the stage 15 μm at a fixed speed, and then hold the stage fixed and measure the relaxation of the force imparted on the microsphere by the sample. We then fit the measured force relaxation curves to a sum of three exponentials (as done in Refs 36,38,39,45) to determine the corresponding relaxation timescales.

We now include a new **SI Fig S2** that plots the measured relaxation times for all three of the systems we study here. The caption describes the process and rationale for measuring the relaxation times. As shown in **SI Fig S2**, the longest relaxation time (τ_3 for the 0.65:0 system) is ~ 3 s. We have added this information to the manuscript as follows:

'Specifically, we set Δt_R to be comparable to the longest measurable relaxation time of the systems (see SI Fig S2). Below this timescale the system would still be relaxing at the onset of the next strain in the cycle, complicating the resulting dynamics, while longer cessation times would unnecessarily prolong measurements, thereby reducing the number of cycles possible before the photobleaching becomes prohibitive. We determine this relaxation time by performing single sweep measurements with 60 s cessation periods and fitting the force relaxation curves to a sum of exponentials, as described in SI Fig S2 and Refs 36,38,39,45.'

'To shed further light on the coupling between the force response, dynamics, and relaxation of the entangled polymers, we evaluate the force relaxation during each $\Delta t_R = 3$ s cessation period between cycles, chosen to be comparable to the longest measurable relaxation timescale of the systems (SI Fig S2).'

'Each oscillatory shear persists for 50 s, including cessation periods of $\Delta t_R = 3$ s between each sweep to allow the network to relax (Fig 1b, SI Fig S2), such that each independent measurement includes 6 – 9 strain cycles (depending on $\dot{\gamma}$) (Fig 1).'

5. - pg4: How is the density of labeling chosen? How does one determine a good signal?

We thank the Reviewer for the question that will help users. We optimize the density of DNA labels such that they are dense enough for ample statistics in small ROIs (128×128 square-pixel ($16.6 \mu\text{m}$)²), while maintaining visible polymer fluctuations and sufficient signal-to-noise. Fig 1c shows an example of a good signal. We use small ROIs in our analysis to achieve sufficient spatial resolution to be able to map the alignment and dynamical scaling as a function of distance from the strain site. The upper limit on labeling density is set by the requirement that polymer fluctuations need to be visible, and that signal from out-of-focus planes do not overwhelm the signal. We have added this information to the Methods section as follows:

'We optimize the density of DNA labels such that they are dense enough for ample statistics in the small ROIs necessary to achieve sufficient spatial resolution, while dilute enough such that polymer fluctuations are resolvable and the imaging is not overwhelmed by signal from out-of-focus planes. Fig 1c shows an example of an optimized DNA labeling density.'

'The concentration of labeled DNA is optimized for DDM measurements as described below and depicted in Fig 1c.'

6. - Fig 1c: why did you choose this specific delta t for the D matrix? Should users be looking for something specific?

We thank the Reviewer for requesting clarification of this choice. We first note that the 2D plots in Fig 1d,e (newly labeled to avoid confusion), are simply to serve as an example of the different degrees of anisotropy of the image structure functions that can be measured at different distances from the strain. We chose a lag time of $\Delta t = 0.25$ s for this example, as at this Δt there are still sufficient correlations between image differences such that a strong signal can be detected. In other words, there are more q -values with warm colors (higher correlations), as indicated by the colorscale that we have added to Fig 1d,e. To determine the alignment factors A_F that we present in Fig 2, we average over lag times $\Delta t = 0.17 - 1$ s where we observe minimal dependence on Δt . $\Delta t = 0.25$ s is within this range.

When examining Fig 1d,e readers should notice that close to the strain $D(q_x, q_y, \Delta t)$ is anisotropic, with two lobes that lie along the axis of the strain path. Conversely, far from the strain path $D(q_x, q_y, \Delta t)$ is isotropic and radially symmetric, as one expects for dynamics with no preferred direction. We have also revised the 2D plots such that each is normalized by its maximum $D(q_x, q_y, \Delta t)$ value (D_{max}). The updated plots show that far from the strain there is less variation in $D(q_x, q_y, \Delta t)$ values, indicative of slower dynamics (less decorrelation). We have clarified this information in the caption of Fig 1 and the main text as follows:

(d,e) $D(q_x, q_y)$ at a representative lag time $\Delta t = 0.25$ s for boxed-in ROIs show that, **(d)** near the strain ($y = 8 \mu\text{m}$), $D(\vec{q}, \Delta t)$ is anisotropic with two lobes that lie along the axis of the strain path, while **(e)** far from the strain ($y = 27 \mu\text{m}$), $D(\vec{q}, \Delta t)$ is isotropic and radially symmetric. Moreover, far from the strain there is less variation in $D(q_x, q_y, \Delta t)$ values compared to close to the strain (i.e., smaller range of colors), indicative of slower dynamics (less decorrelation at the chosen lag time).

7. - pg6: How are beta and delta related? They seem like two independent parameters describing the same types of behaviors in f or τ . For instance, could one have $\beta = 1$, ballistic, but $\delta = 0.5$, super diffusive in the same fit? I realize these parameters were not introduced for the first time in this paper, so details aren't necessary, but some discussion would be helpful.

We thank the Reviewer for the insightful question. The reviewer is correct that β and δ provide similar information, and, in general, their trends should correlate. We generally expect subdiffusion ($\beta > 2$) to pair with stretched exponentials ($\delta < 1$), and superdiffusion or ballistic scaling ($\beta \rightarrow 1$) to pair with compressed exponentials ($\delta > 1$). However, there are cases in which this correlation is not strictly followed, providing important insight into the physical mechanisms underlying the dynamics that would not be possible by only evaluating one or the other. We provide a discussion of these points in the manuscript as follows:

Importantly, we note that while the trends we observe for δ and β are generally consistent with one another and corroborate our interpretations of the data, there are key differences that highlight the need to analyze both metrics to fully capture the dynamics. One important example can be seen by comparing the 0.65:0 data in Fig 5a with that of Fig 4e. Near the strain (small y , red data), β_1 is the most superdiffusive while the corresponding δ is the smallest (closer to 1) for this system compared to 0.5:0 and 0.5:6.4. While the former observation indicates the most pronounced strain-induced directed transport and deformation, the latter indicates the least degree of active transport (or most pronounced confinement) among the three systems – seeming contradictions. Moreover, far from the strain, δ values for the 0.65:0 system are lower ($\delta \approx 0.5$) and exhibit much weaker dependence on q than the other two systems. Taken together, these features suggest that the 0.65:0 system experiences the greatest degree of confinement across all lengthscales (above and below d_T) and y distances from the strain, a feature that is not readily obvious in our analysis of $\tau(q)$ and β_1 and β_2 .

We recognize these differences as indicators of the increased threading probability in 0.65:0 compared to the other systems, in particular the 0.5:6.4 system. As we describe above, threading contributes to the dynamics at all measured lengthscales, whereas entanglements only contribute at $\lambda > d_T$, suggesting that the subdiffusion and confinement that we measure for the 0.65:0 system, in particular, is dominated by threading events. In support of this physical picture, we previously showed that actin-microtubule composites exhibited δ values of ~ 0.5 to 0.7 with the lowest and highest values arising in composites that are highly crosslinked and entangled (no crosslinks), respectively⁶⁸. The observation that the quiescent (large y) δ values for the 0.65:0 system are closer to those reported for crosslinked systems, which have extremely slow relaxation timescales compared to entangled systems, suggests that threaded rings, which also relax much more slowly than entangled rings, play a principal role in the dynamics. This fact, coupled with the comparatively lower δ values near the strain, indicate that the strain-aligned superdiffusion we measure is likely due to entropic stretching and deformation of confined polymers, rather than center-of-mass strain-aligned motion. Notably, this important distinction requires analysis of both $\beta(q)$ and $\delta(q)$.

8. - Fig2: The A_F is a very small number, can the authors give some feeling for what that means in terms of alignment? Is that like % aligned vs random?

We thank the Reviewer for raising this useful question. We agree that the alignment factor A_F is a small number relative to 1, but translating this number to an exact physical quantity, e.g., % aligned vs random, is not straightforward due to a number of conflating factors.

Firstly, we note that previous investigations of anisotropic dynamics and structure have quantified the asymmetry of their data in reciprocal space (i.e., in the q_x - q_y plane) using this alignment factor calculated in the same manner as we do here (e.g., Refs 70,105). In all such studies of which we are aware, the authors compare the values of A_F across different samples or conditions (e.g., shear rate), but do not use A_F to quantify the fraction of objects aligned in a given direction. Quantifying this fraction from A_F would only be reliable in combination with another method to measure the fraction of aligned molecules which could then be used to calibrate A_F values.

Secondly, when considering the low magnitude of A_F it is important to keep in mind the signal to background ratio of our DDM data. As described in the Methods, the DDM image structure function $D(q, \Delta t)$ takes on values that range from B (the background term) to $A + B$ (the amplitude plus background). An advantage of DDM, and one of the factors that motivated us to use the technique, is that dynamics can still be quantified even if the signal to background ratio (A/B) is small. For example, in DOI: 10.1103/PhysRevE.92.042712 the diffusive dynamics of weakly scattering protein clusters were measured with DDM with a signal to background ratio $A/B < 0.1$. For most of our data, we find that $A/B \approx 2$. Given this ratio, when computing A_F from the equation shown in Methods and Fig 1f, we expect to find small values. Specifically, we can approximate the maximum of A_F as $A / (A + 2B)$, so we expect $A_F \lesssim 0.5$, even if the molecules perfectly aligned with the strain. However, as stated above, our approach (following previous approaches to characterize alignment) is to make comparisons of A_F across different conditions, distances from strain, and strain rates. Despite the smallness of A_F , we can see clear statistically significant trends in the values of A_F with these parameters (see Fig 2).

Thirdly, for our experiments, there is a 3 second cessation period in between each strain cycle, as we described in the original manuscript. However, when computing the DDM matrix and A_F , we average over the full length of the movie recorded during the set of 6-9 cycles. Therefore, since the videos contain periods of strain and cessation, the A_F values we find may be smaller than if we only evaluated the strain periods. While one could evaluate A_F only for the strain periods of each 50 s measurement, it would unnecessarily complicate the analysis making our approach more laborious and less user-friendly.

Nevertheless, to demonstrate the effect of averaging over the entire measurement time, we have included a **new SI Fig S3** which shows the instantaneous A_F values for the duration of the cyclic strain for the lowest, highest and resonance strain rate. As shown the peak values are $\approx 3\times$ larger than the average, indicated by the dashed lines, and follow the same non-monotonic trend with strain rate as the average values.

We have added clarifying statements to the text to address this comment as follows:

'Here, θ is defined relative to the x -axis such that isotropic and completely x -aligned dynamics correspond to $A_F = 0$ and $A_F = A(q)/(A(q) + 2B(q))$, respectively, where $A(q)$ and $B(q)$ are amplitude and background terms described below. Increasing values indicate more alignment. The ratio A/B is a measure of the signal-to-noise of the system, such that when this ratio is high (i.e., $A \gg B$), $A_F \rightarrow 1$ for complete x -alignment. For our data, $A/B \lesssim 2$ such that $A_F \lesssim 0.5$.'

'We note that for all DDM analyses, we evaluate the metrics over the entire time of the cyclic strain, which includes both strain and cessation periods. While one could evaluate the strain and cessation periods separately, the analysis is much more laborious and less user-friendly. We choose to not overcomplicate the analysis in this way to facilitate other researchers in adopting our approach. Moreover, we expect the trends we measure to be largely insensitive to our averaging approach, as indicated by the statistically significant trends we present as well as the time-dependent A_F curves shown in SI Fig S3.'

9. - pg 8-10 and fig 3: It is a bit confusing to call affine motion near applied deformation as super diffusive or subdiffusive. Is there any way to relate this language or make an analogy to the usual description continuum description of the expected deformation field in an elastic/viscous continua like $1/r^n$. The more I think about this the more confusing it gets, because the term diffusive suggests random motion, which is almost the opposite of affine. Can the authors comment on this and explain this disconnect between random diffusion and affine deformation? Presumably this has something to do with the coarse graining inherent in DDM.

We thank the Reviewer for requesting clarification on this important point. The distinction, which we now make more clear, when discussing affine motion and diffusive motion is that the former is referring to the collective motion of the network while the latter is referring to the molecules comprising the network. While each molecule is undergoing random thermal fluctuations, the applied strain adds a directed component to the motion. The molecules are stretched and deformed along the strain path but continue to undergo thermal fluctuations. Further, because we analyze the ensemble-averaged dynamics within each ROI that comprises an ensemble of molecules that may have different dynamic characteristics (the coarse graining the reviewer is referring to), the dynamics we measure will have contributions from random thermal motion and strain-induced deformation and motion. Because we are not mapping the displacement field, rather we are quantifying the alignment and scaling of the dynamics of the polymers, we do not relate our profiles to displacement fields for elastic media.

Nevertheless, to avoid confusion on this point, we have removed the word affine from our descriptions of our results. We have also rewritten many sections of the paper to be more careful and clearer in our description of the dynamics and the underlying mechanisms driving the strain alignment and diffusivity.

10. - p11: Beta_1 is like some metric for strain it seems because it is sensitive to larger length scales, whereas Beta_2 is for smaller length scales where only microscopic non affine motion is occurring? Is this correct? Any comment is welcome.

The reviewer is correct that the length scales over which β_1 and β_2 scaling dominate the dynamics are indeed different, with the former being much more sensitive to the applied disturbance than the latter. Specifically, β_1 and β_2 scaling exponents describe the dynamics for lengthscales above and below $\lambda_c \approx$

1.8 μm (i.e., below and above $q_c = 2\pi/\lambda_c$), respectively. As we describe in the manuscript, this crossover lengthscale is comparable to the theoretically predicted tube diameter d_T , which dictates the onset of network confinement and connectivity. Below this lengthscale polymer segments do not feel the effects of the entangling polymers which serve to restrict diffusion far from the strain and, near the strain, pull on neighboring segments in the direction of the strain. The net result is that β_2 is less subdiffusive than β_1 far from the strain path (i.e., $\beta_2 < \beta_1$ for large y values), and lacks the superdiffusive scaling that β_1 exhibits near the strain (i.e., $\beta_2 > \beta_1$ for small y). We have rewritten much of the text to make these points more clear. Example excerpts include:

'As shown in Fig 3a,b, $\tau(q)$ curves for all y values collapse to a single power-law curve, $\tau(q) \sim q^{-\beta_2}$, for $q \gtrsim 4 \mu\text{m}^{-1}$, with a weakly subdiffusive scaling exponent of $\beta_2 \approx 2.6$, suggestive of both weak confinement and minimal strain sensitivity. However, a distinct scaling regime, $\tau(q) \sim q^{-\beta_1}$, emerges for smaller q values with a stark monotonic decrease in β_1 as y decreases (cool to warm colors).'

'...we recognize that the wavevector q_c at which scaling crosses over from β_1 to β_2 corresponds to a lengthscale $\lambda_c = 2\pi/q_c \approx 1.6 \mu\text{m}$, which is remarkably close to the predicted tube diameter of the 0.65:0 DNA solution, $d_{T,0.65} \approx 1.5 \mu\text{m}$ (see Methods). At lengthscales smaller than d_T , corresponding to $q > q_c$ and β_2 scaling, DNA segments do not feel the surrounding entanglements so there is both minimal tube confinement (thus weaker subdiffusion), and minimal 'pulling' by entanglements that are moving along with the strain (thus no directed motion).'

'To quantify and compare the biphasic dynamic scaling described above for all three systems, we plot their respective β_1 (Fig 3e) and β_2 (Fig 3f) exponents as functions of y . As shown in Fig 2a-c, β_2 values, which represent scaling behavior for lengthscales below λ_c , show minimal y dependence for all systems. Insofar as the y dependence reflects strain-induced dynamics, this result indicates that below the DNA entanglement lengthscale (recall $\lambda_c \approx d_T$) the effects of the strain are screened by the surrounding entanglements. We also point out that all β_2 values are subdiffusive with an average value of $\langle \beta_2 \rangle \approx 2.7$. While subdiffusion, at first glance, seems plausible, as it is typically an indicator of strong confinement, such as in entangled or crowded polymer systems^{33,40,41,80,81}, recall that β_2 values quantify the dynamics for lengthscales below the tube diameter, where we actually expect confinement to have little impact. Rather, the DNA should be undergoing unconfined normal Brownian diffusion.

This effect suggests that there is another confinement mechanism that persists across the entire q range ($0.9 \mu\text{m} < 2\pi/q < 6.3 \mu\text{m}$), even in the absence of microtubules. As we discussed in the Introduction, threading of ring polymers by their linear counterparts can lead to strong subdiffusion and caging as well as suppressed relaxation of both rings and linear chains in ring-linear blends^{44,53,82,83}. Moreover, the effects of threading are most prominent in blends with comparable concentrations of rings and linear chains, as in the systems we study here⁴⁴. While the displacement of a threaded ring parallel to the contour of the threading linear chain is limited to d_T (similar to an entangled polymer) the lateral displacement of the ring is confined to its radius of gyration $R_{G,R}$ ($\sim 0.6 \mu\text{m}$ for the DNA in our systems⁸⁴). As such confinement from threading impacts dynamics for lengthscales $\lambda > R_{G,R}$ which correspond to $q \lesssim 10 \mu\text{m}^{-1}$, encompassing our entire q range ($1 \mu\text{m}^{-1} \lesssim q \lesssim 7 \mu\text{m}^{-1}$)⁸⁵. As such, threading is likely the dominant mechanism mediating the small-lengthscale ($q > q_c$, $\lambda < \lambda_c$) subdiffusive scaling quantified by β_2 .

'We now turn to understanding the scaling behavior at larger lengthscales, i.e., $\lambda \gtrsim d_T$, where entanglements need be considered: namely β_1 . Far from the strain site (large y), where the effect of the strain is negligible, we expect entanglements to lead to more pronounced subdiffusion, compared to β_2 , as they enhance the degree of confinement (beyond threading). Conversely, near the strain site (small y), where the imposed strain dominates the dynamics, entanglements should increase strain-oriented motion and stretching, as described in the previous section, manifesting as superdiffusive or ballistic

dynamics. This trend is exactly what we observe in Fig 3e, in which β_1 transitions from ~ 1 (ballistic) near the strain site to ~ 3 (subdiffusive) furthest from the strain.

The takeaway here is that at lengthscales smaller than the tube diameter, threading dominates the dynamics for all systems, while entanglements dominate at larger lengthscales. The onset of entanglement dynamics enhances subdiffusion far from the strain and alignment near the strain, with the 0.65:0 blend exhibiting the strongest threading and entanglement effects.'

11. - p11: I am struggling to somehow relate the betas to strain. I am not sure if there is a good relationship here or if the authors can explain. At some point in the manuscript, it would be helpful to explicitly point out what the power of DDM is over simply doing PIV to get a strain field. This may help clarify some confusion and highlight the authors results.

We thank the Reviewer for requesting clarification on this important point, as the use of DDM over PIV is indeed central to our advance. Strain fields that PIV can indeed generate provide the spatially-resolved displacement of the system in response to a disturbance. However, PIV does not provide any information about the dynamics of the macromolecules that generate the strain field. The scaling exponents, β , determined from DDM, as well as δ and A_F , characterize the type of motion that the macromolecules undergo in response to the disturbance – providing much richer information than a simple strain field can provide. We now make this point clear in the manuscript. We also describe other advantages to using DDM over PIV or single-particle tracking (SPT), as follows:

'We use DDM instead of more conventional single-particle tracking (SPT) or particle image velocimetry (PIV) methods to allow for a high density of labeled macromolecules that are smaller, dimmer, and more susceptible to photobleaching as compared to microspheres and other standard probes^{9,10}. The high density, which prevents tracking single particles (as is done in SPT and PIV), is critical to spatially resolving statistically robust dynamics within each ROI; while a lower signal threshold than is needed for SPT and PIV facilitates using DNA or other macromolecules.'

'Importantly, as described above, our implementation of DDM in OpTiDDM enables precise quantification of macromolecular dynamics over 3 decades of length and time scales ($\sim 0.6 \mu\text{m} - 30 \mu\text{m}$, $20 \text{ms} - 100 \text{s}$), well above and below the range that typical implementations of SPT and PIV can measure, particularly for dense or noisy systems. Moreover, our dynamical characterization of the surrounding network—which quantifies macromolecular dynamics, deformations and relaxation profiles in response to strain—is distinct and arguably much richer than mapping a simple strain or displacement field that PIV can generate. Finally, PIV provides a deformation field by assuming constant velocity motion of system between each lag-time. This analysis can provide an estimate of the deformation of the system in response to an applied strain or disturbance. However, in entangled and crowded polymer systems, the assumption of directional ballistic motion between a given time interval is not reliable. DDM analysis, on the other hand makes no assumptions about the dynamics, but rather quantifies the dynamics by evaluating the correlations between image differences. As such, DDM can not only determine the degree of alignment (similar to PIV), but it can elucidate the nature of the motion arising from polymer fluctuations, deformations, re-orientation and flow as described above.'

12. - p11: The details are good, but almost overwhelming, a contextual paragraph before and after would be very welcome

We thank the Reviewer for the suggestion. We have rewritten this section almost entirely to provide more context and eliminate details that are not critical to understanding our central findings. In particular we have modified and added the following paragraphs before and after the discussion of threading:

'This effect suggests that there is another confinement mechanism that persists across the entire q range ($0.9 \mu\text{m} < 2\pi/q < 6.3 \mu\text{m}$), even in the absence of microtubules. As we discussed in the Introduction, threading of ring polymers by their linear counterparts can lead to strong subdiffusion and caging as well as suppressed relaxation of both rings and linear chains in ring-linear blends. Moreover, the effects of threading are most prominent in blends with comparable concentrations of rings and linear chains, as in the systems we study here. While the displacement of a threaded ring parallel to the contour of the threading linear chain is limited to d_T (similar to an entangled polymer) the lateral displacement of the ring is confined to its radius of gyration $R_{G,R}$ ($\sim 0.6 \mu\text{m}$ for the DNA in our systems)⁸⁴. As such confinement from threading impacts dynamics for lengthscales $\lambda > R_{G,R}$ which correspond to $q \lesssim 10 \mu\text{m}^{-1}$, encompassing our entire q range ($1 \mu\text{m}^{-1} \lesssim q \lesssim 7 \mu\text{m}^{-1}$)⁸⁵. As such, threading is likely the dominant mechanism mediating the small-lengthscale ($q > q_c$, $\lambda < \lambda_c$) subdiffusive scaling quantified by β_2 .'

'The takeaway here is that at lengthscales smaller than the tube diameter, threading dominates the dynamics for all systems, while entanglements dominate at larger lengthscales. The onset of entanglement dynamics enhances subdiffusion far from the strain and alignment near the strain, with the 0.65:0 blend exhibiting the strongest threading and entanglement effects.'

13. - p15-16: The monotonic force behavior and non-monotonic strain behavior is almost begging for a physical model. Do the authors know of any possible material models to explain this?

We agree with the Reviewer that a physical model would be an excellent contribution. One motivation for publishing this work is in fact to spark theoretical interest in this intriguing behavior. We do not know of any material models to quantitatively describe our results, but we do provide a thorough discussion of our understanding of the behavior based on the properties of the polymers, prior experimental results and models for entangled polymers. We have expanded on this discussion in the revised version of the manuscript. Some relevant excerpts include:

'Comparing the alignment factor $A_{F,i}$ to the maximum force $F_{M,i}$ (Fig 8c) reveals that for slow strain rates, $A_{F,i}$ generally increases with increasing $F_{M,i}$ until $\dot{\gamma} = 42 \text{ s}^{-1}$ followed by a decrease. The non-monotonic dependence of $A_{F,i}$ on $F_{M,i}$ is strongest for 0.65:0, demonstrating maximal strain-aligned deformation and stretching. Also notable is the fact that while $A_{F,i}$ values for 0.5:6.4 are comparable to those for 0.5:0, the corresponding $F_{M,i}$ values are substantially higher, indicating that rigid microtubules require substantially higher forces for similar deformation.

We see similar relations between $\beta_{1,i}$ and $F_{M,i}$ (Fig 8d), with $\beta_{1,i}$ generally decreasing with increasing $F_{M,i}$ until $\dot{\gamma} = 42 \text{ s}^{-1}$ after which $\beta_{1,i}$ increases with $F_{M,i}$ for the DNA-only blends (0.5:0 and 0.65:0) while it remains constant for the DNA-MT composite (0.5:6.4). Further, the 0.65:0 system displays the most extreme non-monotonicity in $\beta_{1,i}$ with increasing $F_{M,i}$, as well as the lowest $\beta_{1,i}$ values, indicating the highest degree of strain-coupling and directed stretching and deformation. This strain alignment, in turn, results in shear-thinning, i.e., reduced increase in $F_{M,i}$ with increasing $\dot{\gamma}$.

How the varying degrees of alignment ($A_{F,i}$) and directed motion ($\beta_{1,i}$) impact the elastic retention of force following the strain ($F_{R,i}$) can be seen in Fig 8e and Fig 8f, respectively. Similar to Fig 8c,d, $A_{F,i}$ and $\beta_{1,i}$ exhibit mirror dependences on $F_{R,i}$ with all systems displaying non-monotonic $\dot{\gamma}$ dependence, with maxima and minima in $A_{F,i}$ and $\beta_{1,i}$, respectively, occurring at $\dot{\gamma} = 42 \text{ s}^{-1}$. Further, the 0.65:0 system displays the largest range in $A_{F,i}(\dot{\gamma})$ and $\beta_{1,i}(\dot{\gamma})$ values, indicating the largest dynamic range and the strongest resonant coupling of relaxation rates and strain rates. Moreover, the 0.65:0 system switches from exhibiting behavior similar to the 0.5:0 blend to that of the 0.5:6 composite. The latter result once again demonstrates a switching from dissipative (small $F_{R,i}$) to elastic-like (large $F_{R,i}$) dynamics for the

0.65:0 blend at $\dot{\gamma} \approx v_e$. In contrast, the other two systems display either dissipative (0.5:0) or elastic-like (0.5:6) behavior across all $\dot{\gamma}$.

Intriguingly, $A_{F,i}$ decreases (and $\beta_{1,i}$ increases) with increasing $F_{R,i}$ for 0.5:0 and 0.5:6.4 whereas the opposite trend is seen for the 0.65:0 system. We can understand this phenomenon as arising from the varying degree to which DNA builds up or is compressed at the leading edge of the probe versus preferentially aligning with the strain path. Namely, reduced alignment (smaller $A_{F,i}$) coupled to increased elastic storage (larger $F_{R,i}$), as seen for 0.5:0 and 0.5:6, indicates that it is polymer buildup at the strain edges, which increases the local entanglement density that, in turn, reduces the ability of the polymers to deform and move in response to the strain (reducing/increasing $A_{F,i}/\beta_{1,i}$). This phenomenon increases the polymer relaxation times, resulting in reduced relaxation and increased ability to elastically maintain induced force (increasing $F_{R,i}$).

On the other hand, the coupled increase of alignment ($A_{F,i}$) and flow ($\beta_{1,i}$) with elasticity ($F_{R,i}$) for 0.65:0 corroborates that enhanced strain-aligned deformation and stretching (increasing/decreasing $A_{F,i}/\beta_{1,i}$) arises from increasingly strong threadings and entanglements which increase the elasticity ($F_{R,i}$). The strong entanglements prevent polymer compression and support entropic stretching, as described above. Namely, polymers are pulled in the direction of the strain from entanglements with 'leading' polymers as well as opposite to the strain from entanglements with 'trailing' polymers. The net result is increased entropic stretching and affine deformation, which, in turn, suppresses buildup at the strain edges and promotes shear-thinning.'

We also point the Reviewer to our response to Comment #15 below for discussion of a non-monotonic dependence of the energy dissipation rate determined from our force measurements.

14. - p16: some of the discussion here about polymer buildup or leaving wakes behind seem like they could be analyzed using the authors technique, by applying DDM at these locations. If this is true, perhaps the authors can comment on how this may be done in the future if they dont want to include it in this study.

We thank the Reviewer for the suggestion and agree that this would be a welcome advance to our platform. We did in fact attempt this on many occasions but were unable to get statistically robust results given the small ROI of the strain path. One issue is that we are using wide-field fluorescence microscopy to image the DNA, so out-of-focus light contributes to the signal, such that any wake or buildup is less obvious. To circumvent this issue, one could use confocal or light-sheet microscopy to reduce out-of-focus signal. There are examples of optical tweezers coupled to these microscopy modalities in the literature that one could follow to build the necessary instrumentation. We have added this information to the revised text as follows:

'Nevertheless, should one require increased signal-to-noise, OpT could be coupled with confocal or light-sheet microscopy that reduce out-of-focus signal¹⁰¹⁻¹⁰⁴. These modalities may also prove useful for visualizing strain-induced polymer clustering and spatial heterogeneities that we do not characterize here.'

15. - p16: Force dissipation is a bit of a confusing term. Assuming the force dissipates over time, then there is a timescale (or velocity) involved and then it is usually discussed as an energy dissipation ($F \cdot v$). Elastic memory is also a confusing term - seems like elastic should be viscoelastic? I dont understand how FR/FM is a measure of relative elastic memory versus dissipation. It just seems like a measure of dissipation alone, which would also depend on the strain rate because $P = F \cdot v$. The lower the FR/FM, the lower the residual force is - meaning the most "force" was dissipated - so it just looks like higher strain

rate leads to more “force” dissipation? — which maybe leads to an equal amount of energy dissipated in $F \cdot v$? Any clarification or comment here is welcome.

We thank the Reviewer for requesting clarity regarding our terminology when discussing the force dissipation. We first clarify what the maximum force F_M and residual force F_R we measure correspond to. Our measurement protocol consists of moving the bead $15 \mu\text{m}$ at a constant speed, waiting 3 s (as we discuss in our response to Comment #4), then moving the bead $15 \mu\text{m}$ in the opposite direction. The maximum force is the force that the system exerts on the bead at the end of each $15 \mu\text{m}$ strain (as the force increases monotonically during the strain). The residual force is the force that the system continues to exert on the bead at the end of the 3 s cessation period. This cessation period is chosen to be longer than the longest measurable relaxation time of the system (see SI Fig S2), such that F_R can be assumed to indicate the elastic contribution to the stress response.

If an elastic material is strained and then held at that strain (as during the cessation periods in our experiments), the imposed stress (force) will remain until the strain is released. For a system that is purely viscous (i.e., dissipative), the residual force would be zero as the force is proportional to speed (i.e., Stokes’ drag). In other words, once the bead stops moving (or the applied strain remains fixed), as at the onset of each cessation period, any force quickly decays to zero. As such, the quantity F_R/F_M is a measure of the relative elasticity (i.e., storage) of the material—the fraction of the induced force F_M that remains following the network relaxation. The Reviewer is correct that it is not entirely accurate to describe it as measuring the relative elastic memory versus dissipation. We have changed the description as follows:

The fractional force that is sustained during Δt_R , which we define as F_R/F_M (Fig 7b), is a measure of the relative elastic storage of the system. For reference, a completely viscous, dissipative system immediately relaxes all induced stress once the straining stops (i.e., the bead stops moving), in analogy with Stokes drag, such that $F_R/F_M \rightarrow 0$ and the polymers store no memory of their initial pre-strained state. Conversely, a purely elastic system stores the induced stress for as long as the applied strain is sustained (the bead remains fixed at the maximal strain position), maintaining memory of its pre-strained state, such that $F_R/F_M \rightarrow 1$.

Regarding our use of *dissipative*: We chose the term dissipation to refer to the relaxation of force following an imposed strain, as the relaxation arises from the polymers moving, rearranging and redistributing to alleviate stress: dissipative processes. Any energy input from the strain dissipates to thermal energy via macromolecular fluctuations, such that the force relaxes to zero (without an elastic component). However, because we are measuring force rather than energy, to avoid confusion, we have changed our terminology through the paper to refer to force relaxation rather than force dissipation.

Regarding our use of *elastic memory*: As we describe in our response to Comment #1, what we mean by elastic memory is the extent to which the resistive force the network exerts once the bead stops moving does not relax. Rather, the material sustains the force such that it has memory of its initial pre-strain state, stored as elastic potential energy, so it can return to this state. We have used this terminology in previous works to describe the extent to which systems are able to relax induced stress, thereby erasing any memory of the previous state; or, alternatively, retain the stress, and thus have ‘memory’ of its previous state. However, to avoid confusion we have replaced many references to elastic memory with elasticity or elastic storage. We have also added the clarifying statement reproduced above to the manuscript.

Finally, we appreciate the reviewer's suggestion to evaluate $F_R \dot{\gamma}$ to connect our force measurements to the energy dissipation rate for viscoelastic materials. Firstly, we clarify that we measure F_R for the period of the measurement in which no straining is applied, so it is an indicator of the force that remains in the system once dissipative processes to relax imposed stresses have ceased. This value is primarily independent of the strain rate, as shown in Fig 7a, indicating that it is an intrinsic property of the system, rather than a strain-dependent term.

As the reviewer notes, F_R/F_M , on the other hand, decreases with increasing strain rate. If this trend is due to the criterion that an equal amount of energy is dissipated, as the reviewer suggests, then $\dot{\gamma} * F_R/F_M$ should be independent of $\dot{\gamma}$. However, as we show in our **new inset to Fig 7b** (reproduced to the right) in which we plot $\langle F_R/F_M \rangle \dot{\gamma}$ versus $\dot{\gamma}$, there is a clear $\dot{\gamma}$ dependence for all systems. The larger the quantity $\langle F_R/F_M \rangle \dot{\gamma}$, the lower the rate of energy dissipation, indicating more energy is being stored elastically.

As shown in Fig 7b, $\langle F_R/F_M \rangle \dot{\gamma}$ increase monotonically with similar scaling for the 0.5:0 and 0.5:6.4 systems, with the magnitude of the DNA-MT composite (0.5:6.4) being ~2-3 fold higher than the corresponding DNA-only system (0.5:0). These trends are in line with the physical picture of polymer buildup at the strain edges that is more pronounced at slower speeds. As the strain rate increases, the deformation energy goes increasingly into elastically stretching the polymers rather than the dissipative processes of moving and rearranging the polymers into high- and low-density regions. The higher magnitude for the DNA-MT composite is an indicator of the increased elastic storage that the stiff microtubule network provides, which is independent of $\dot{\gamma}$.

Contrastingly, the 0.65:0 system displays the non-monotonic 'resonant' dependence that the corresponding DDM data shows. Namely, $\langle F_R/F_M \rangle \dot{\gamma}$ is maximized at $\dot{\gamma} = 42 \text{ s}^{-1}$, indicating that the rate of energy dissipation is minimized. At this maximum, $\langle F_R/F_M \rangle \dot{\gamma}$ is slightly larger than the DNA-MT composite (albeit within error), whereas at the slowest strain rate, $\langle F_R/F_M \rangle \dot{\gamma}$ for 0.65:0 is slightly lower than the 0.5:0 system.

These results shed further light on the physical mechanisms that give rise to the dynamics and force response we measure. Firstly, they demonstrate that the processes underlying the force response and dynamics of the 0.65:0 system are distinct from the 0.5 mg/ml DNA systems. In the former case, entropic stretching and flow alignment, with minimal dissipative rearrangement, dictate the dynamics and mechanics; and is most pronounced when the strain rate is resonant with the fastest polymer relaxation rate. In the latter case, the data is suggestive of polymer buildup at the leading edges of the probe that arises from reduced network connectivity and strain-coupling. Moreover, these data show that at lower strain rates, the DNA in the 0.65:0 system can move and rearrange to dissipate energy, similar to the 0.5:0 system, whereas at the resonant rate, the 0.65:0 system deforms more elastically than the network of stiff MTs, which manifests as entropic stretching and flow alignment.

We have added this plot to Fig 7b, and have added the description above to the main text where the figure is discussed.

16. pg 18: The statement about what slower and fast strain rates lead to could use a reference if this is known in the literature. If not, and it is conjecture, this should also be made clear.

We thank the Reviewer for the suggestion. We have added appropriate references and rewritten the paragraph to make clear what is known from the literature and what is conjecture. The paragraph now reads:

'This process may also explain the $\dot{\gamma}$ dependence of F_R/F_M for 0.5:0 and 0.5:6.4. Namely, slower strains would provide more time for the polymers to rearrange and untangle with neighbors to stay at the leading edge of the probe, thereby increasing F_R/F_M ; whereas faster strain rates would result in more polymers slipping off of the moving probe as they are pulled back by entangling polymers, thus decreasing F_R/F_M .^{38,66}

Unlike 0.5:0 and 0.5:6.4, the 0.65:0 system displays minimal dependence on $\dot{\gamma}$ or cycle, suggestive of a different mechanism dictating the local force response; namely, $\dot{\gamma}$ -dependent shear thinning and flow alignment facilitated by threading (as described above)^{38,76,87}. Specifically, we rationalize that enhanced threading and preferential alignment with the strain path limits the extent to which polymers buildup at the edges of the strain path, thereby negating any dependence on time (cycle number). Likewise, the shear-thinning that manifests from the strain alignment results in less increase in F_R/F_M as $\dot{\gamma}$ increases, due to the lower local entanglement density arising from polymer stretching and deformation along the strain path.'

17. pg 21: is there a way to generalize the observations in these DNA blends to general polymer systems? And perhaps clear implications of these findings?

We thank the Reviewer for the suggestion. We have added statements to the Conclusions section to discuss the general applicability of our results to polymer systems and the clear implications of our findings, as follows:

'Further, wherever possible, we position our discussion of physical mechanisms driving our results in the context of general polymer physics theory and rheology principles, such that our results can be generalized to other polymeric systems and users can draw from our arguments to make sense of their own results on other polymeric systems. Finally, it is important to note that DNA has been used as a model polymer system for decades now to elucidate open questions in polymer physics^{26,89-92}. As such, our results using DNA are generally transferrable to polymer systems with similar number of entanglements per chain. Notably, our systems do include rings, and our results suggest that threading plays an important role in the dynamics, stress response and propagation. For example, more pervasive threading in the 0.65:0 system as compared to the other two systems, is the likely mechanism that drives increased confinement and subdiffusion at lengthscales below that of the entanglement tube confinement, as well as strain-aligned deformation and stretching that, in turn, facilitates shear-thinning in response to the local disturbance. These important findings are directly applicable to investigations of ring-linear blends which continue to be the topic of fervent research and debate^{33,35,36,38,76,86,87,93-97}.

'We rationalize our results as arising from the varying propensity for polymers to build up at the strain edges versus align and entropically stretch along the strain path, likely dictated by the coupled effects of polymer entanglements, threadings, and the varying relaxation rates and mechanisms accessible to flexible DNA versus rigid microtubules. Importantly, our observations and interpretations of the intriguing physical phenomena that we present here require the direct coupling of macromolecular dynamics and deformation fields to stress response—made uniquely possible by coupling OpT microrheology with DDM.'

18. pg 23: are DDM measurements and force measurements done simultaneously?

No they are not. As we describe in our response to Comment #2, we have revised the text to make this point clearer. We have added the following clarifying statement to the Methods where the reviewer indicated:

'For DDM measurements, we use a piezoelectric actuator mirror (PI USA) to move the trap relative to the sample chamber while keeping the 600×900 square-pixel (130 nm/pixel) field-of-view (FOV) of the camera fixed and centered at the resting trap position. For force measurements, performed independently of DDM measurements, we use a piezoelectric nanopositioning stage (Mad City Laboratories) to move the sample relative to the fixed trap.'

19. pg 23: Explain how $\Delta t = 3s$ was chosen.

We thank the Reviewer for the suggestion, which we address in our response to Comment #4. We have added this information to the Methods section where the Reviewer indicated, as follows:

Specifically, we set Δt_R to be comparable to the longest measurable relaxation time of the systems (SI Fig S2). Below this timescale, the system would still be relaxing at the onset of the next strain in the cycle, complicating the resulting dynamics. Longer cessation times, on the other hand, would unnecessarily prolong measurements, thereby reducing the number of cycles possible before the photobleaching becomes prohibitive. We determine this relaxation time by performing single sweep measurements with 60 s cessation periods and fitting the force relaxation curves to a sum of exponentials, as described in SI Fig S2 and Refs 36,38,39,45.

20. pg 24: Can any interpretation for the value of $A_F = 0.02$ with radial averaging be provided? The number is very low (maybe lower due to radial averaging), so can the authors say a bit more about what they think this means?

We thank the Reviewer for requesting clarification on this point. The alignment factor A_F is not actually determined from the radially averaged image structure function. Rather, we compute A_F from the 2D image structure function $D_i(q_x, q_y, \Delta t)$ where q_x and q_y are x and y components of the wave vector \vec{q} (no radial averaging). We then compute A_F in q -space via $A_F(|\vec{q}|, \Delta t) = \int_0^{2\pi} D(q, \Delta t, \theta) \cos(2\theta) d\theta / \int_0^{2\pi} D(q, \Delta t, \theta) d\theta$, where $q = |\vec{q}|$ and θ is defined relative to the x -axis.

Independent of our A_F analysis, to determine the type and rate of DNA motion, we radially average $D(\vec{q}, \Delta t)$ to get an image structure function that can be described by $D(q, \Delta t) = A(q)[1 - f(q, \Delta t)] + B(q)$, where $f(q, \Delta t)$ is the intermediate scattering function (ISF), $A(q)$ is the amplitude, and $B(q)$ is the background. By fitting the ISF to a stretched or compressed exponential, $f(q, \Delta t) = e^{-(\Delta t/\tau(q))^\delta}$, we determine the correlation decay time $\tau(q)$ and scaling exponent δ that describe the dynamics. We then evaluate the power-law scaling of the decay time $\tau(q) \sim q^{-\beta}$ to describe the type of motion, with $\beta = 2, > 2, < 2$ and 1 indicating diffusion, subdiffusion, superdiffusion and ballistic transport, respectively.

As we describe in the Methods section (which is likely the source of confusion), radial averaging is only strictly accurate when the image structure function is radially symmetric, in which case $A_F = 0$, which is not the case for our data near the strain. As such, we provide the caveat that the radial symmetry is an approximation. However, as we note in the manuscript, the fact that A_F in the most aligned case is relatively small (~ 0.02), as the reviewer points out, provides justification for radial averaging. We have clarified this section as follows:

'To determine the extent to which the DNA dynamics are preferentially aligned along the strain path (x -axis) (Fig 2), we adopt alignment factor analysis typically used to analyze scattering data produced by an

aligned field^{70,105}. We compute an alignment factor A_F with respect to the strain path (x -axis) by computing weighted azimuthal integrals of the 2D image structure function $D(q_x, q_y, \Delta t)$ (i.e., integrals over θ where $\theta = \tan^{-1}(q_y/q_x)$) where q_x and q_y are x and y components of the wave vector \vec{q} : $A_F(q, \Delta t) = \int_0^{2\pi} D(q, \Delta t, \theta) \cos(2\theta) d\theta / \int_0^{2\pi} D(q, \Delta t, \theta) d\theta$ (Fig 1f). Here, θ is defined relative to the x -axis such that isotropic and completely x -aligned dynamics correspond to $A_F = 0$ and $A_F = A(q)/(A(q) + 2B(q))$, respectively, where $A(q)$ and $B(q)$ are amplitude and background terms described below. Increasing values of A_F indicate more alignment. The ratio A/B is a measure of the signal-to-noise of the system, such that when this ratio is high (i.e., $A \gg B$), $A_F \rightarrow 1$ for complete x -alignment. For our data, $A/B \lesssim 2$ such that $A_F \lesssim 0.5$. To obtain a single A_F value for each distance y , we average over $\Delta t = 0.17$ - 1 s and $q = 1$ - $7 \mu\text{m}^{-1}$, where there is no statistically significant dependence of A_F on these parameters.'

For an explanation of the relatively small values of A_F , we direct the reviewer to our Response to their Comment #8.

Beyond our Response to Comment #8, we also point out that while our measured A_F values are small relative to the maximum $A_F \approx 0.5$ for our setup, the variation in values for different y distances and strain rates are significantly larger than the error, so the trends we measure can be considered robust and markers of variations in the alignment of the DNA motion. Moreover, we have a lower limit on ROI size of $(16 \mu\text{m})^2$ to ensure ample statistics, so even our measurement closest to the strain includes molecules that are ~ 10 tube diameters ($d_T \approx 1.6 \mu\text{m}$) from the strain, so likely are much less aligned than those ~ 1 - $2d_T$ from the strain.

20. pg 24, If I am interpreting the parameters in the ISF correctly then you end up with an $\exp(-\Delta t / q^\beta)$, so then there are two competing parameters (β and Δt) that are multiplied here. But it seems possible to determine them independently, can the authors comment on this point?

We thank the Reviewer for requesting clarification of this point, which we partially address in our response to Comment #20. In fact, we do determine β_1 and β_2 independently of determining δ . Specifically, by fitting the ISF to a stretched or compressed exponential, $f(q, \Delta t) = e^{-(\Delta t / \tau(q))^\delta}$, we determine a correlation decay time τ and scaling exponent δ for every q value. We make no assumptions about the function form of $\tau(q)$. We then evaluate the power-law scaling of $\tau(q)$ to determine β_1 and β_2 . However, we do not a priori assume power-law scaling $\tau(q) \sim q^{-\beta}$, and, in fact, our data do not follow this simple scaling but instead shows two distinct scaling regimes for small and large q values. We have edited the text to make this distinction clearer:

'The second metric is the density fluctuation decay time $\tau(q)$ (Fig 1g), determined by fitting the intermediate scattering function (ISF) determined from $D(q, \Delta t)$, which describes the type of motion, e.g., ballistic, superdiffusive, diffusive, subdiffusive, etc^{9,10}. Specifically, the scaling exponent associated with the power-law dependence of $\tau(q)$, i.e., $\tau(q) \sim q^{-\beta}$, provides this characterization, with $\beta = 1$, $\beta = 2$, $1 < \beta < 2$, and $\beta > 2$ corresponding to ballistic, diffusive, superdiffusive and subdiffusive dynamics, respectively.

The third metric is the exponent δ that stretches or compresses the exponential function that describes the ISF (see Methods). Dilute and isotropic thermal systems undergoing diffusive (Brownian) motion exhibit $\delta = 1$ scaling. However, superdiffusive or ballistic-like motion, as seen in actively driven systems, often manifest $\delta > 1$,⁷²⁻⁷⁴ and subdiffusive dynamics that crowded and heterogeneous systems exhibit are typically better fit to $\delta < 1$.^{23,25,68,71,75}

'By evaluating the functional form of $\tau(q)$ determined from fitting the ISF, we analyze the extent to which $\tau(q)$ can be described by power-law scaling $\tau(q) \sim q^{-\beta}$ where the scaling exponent β describes the type of motion. Specifically, $\beta = 2$ is indicative of normal Brownian diffusion whereas $\beta \rightarrow 1$ describes superdiffusive or ballistic dynamics and $\beta > 2$ indicates anomalous subdiffusion^{10,24}.'

Reviewer #3 (Remarks to the Author):

The authors perform experiments in which they combine two experimental tools that they had previously employed (Optical Tweezers and Differential Dynamic Microscopy) to map the response to local nonlinear strains in polymeric fluids, namely entangled DNA solutions with and without incorporated microtubules.

Overall, I find that the manuscript has good potential and could eventually be published in Nature communications. The methodology presented is undoubtedly innovative and intriguing, and some of the results obtained appear interesting and promising: among other things, the main result seems to be a synergic response of the DNA blends to perturbations with strain rates of the order of the intrinsic entanglement rate, an effect that seems to be suppressed by incorporation of microtubules.

We are pleased that the Reviewer finds our work 'innovative and intriguing' and our results 'interesting and promising'. We are equally pleased that the Reviewer acknowledges that our 'manuscript has good potential' to 'be published in Nature Communications'.

The present version of the manuscript let the reader clearly understand that the authors did a great amount of work, which seems to be conducted in an appropriate way. Unfortunately, the authors fail to offer a clear description of the main mechanisms behind all their observations, not being true to their promise in the abstract to "unambiguously connect polymer dynamics to force response".

The main problem is that the authors mostly list a large number of experimental findings without providing a clear pathway to their interpretation and leaving many questions unanswered.

In addition, I find the current version very little appealing, I find particularly disappointing the lack of accent in the storytelling, which carries the reader for 24 pages without highlighting in a proper way the key results and the crucial steps in getting such results.

Maybe, the authors should consider moving some of the discussion out of the main text, by keeping only those parts that constitute the core message of the paper.

Having read the manuscript three times, I am still unable to provide a concise list of changes because the problem with the present manuscript cannot be solved locally but requires a global rewriting that make it fit a multidisciplinary journal, publishing in which requires an effort to present facts and hypotheses in the most possible clear way, by clearly outlining the noteworthy results, and why are they significant in the authors' opinion. Both goals are not reached by the current version.

We thank the Reviewer for the suggested rewrite to more clearly emphasize the key results, underlying mechanisms, and their implications. We agree with the reviewer that the original version of the manuscript was dense and provided a lot of detailed information. We chose this approach because one of the main results of the paper is, in fact, the presentation of the OpTiDDM technique itself, we feel it is important to keep a lot of the detail describing the measurements and the metrics that can be extracted using our approach. Moreover, only after presenting the DDM and the OpT results, is the reader ready to understand the meaning of the results because it requires the combination of the two measurements. As such, we did not provide a lot of interpretation and mechanistic discussion until the end of the manuscript.

However, in light of the Reviewer's useful suggestions we have substantially rewritten the Results and Discussion section of the manuscript, as the Reviewer will see by all the blue font text in the manuscript. We now provide more contextualization and emphasis of our key results, as well as more descriptions of the physical mechanisms that underlie our results. We have also removed and/or truncated details and findings that are not critical to understanding our most important findings. Our substantive re-write has

made our work more accessible, digestible and of broader interest to the interdisciplinary readership of Nature Communications. Below we include a few excerpts that the reviewer may find helpful. Although a full read-through is necessary to capture the global rewrite.

'We focus our DDM analysis on three metrics that can be extracted from the videos by analyzing their image differences in Fourier space and computing an image structure function $D(\vec{q}, \Delta t)$ for varying wavevectors q of each pair of image differences separated by a lag time Δt (see Fig 1c-g, Methods)^{9,10,24,69}. The first metric we examine is the alignment factor A_F (see Fig 1f, Methods)⁷⁰, to determine the degree to which polymer motion is preferentially aligned along the strain direction versus randomly distributed. Increasing fractional values of A_F correlate with increasing alignment with $A_F = 0$ indicating completely isotropic motion.

The second metric is the density fluctuation decay time $\tau(q)$ (Fig 1g), determined by fitting the intermediate scattering function (ISF) determined from $D(q, \Delta t)$, which describes the type of motion, e.g., ballistic, superdiffusive, diffusive, subdiffusive, etc^{9,10}. Specifically, the scaling exponent associated with the power-law dependence of $\tau(q)$, i.e., $\tau(q) \sim q^{-\beta}$, provides this characterization, with $\beta = 1$, $\beta = 2$, $1 < \beta < 2$, and $\beta > 2$ corresponding to ballistic, diffusive, superdiffusive and subdiffusive dynamics, respectively.

The third metric is the exponent δ that stretches or compresses the exponential function that describes the ISF (see Methods). Dilute and isotropic thermal systems undergoing diffusive (Brownian) motion exhibit $\delta = 1$ scaling^{9,71}. However, superdiffusive or ballistic-like motion, as seen in actively driven systems, often manifest $\delta > 1$,⁷²⁻⁷⁴ and subdiffusive dynamics that crowded and heterogeneous systems exhibit are typically better fit to $\delta < 1$.^{23,25,68,73,75}

Importantly, our implementation of DDM in OpTiDDM enables precise quantification of macromolecular dynamics over 3 decades of length and time scales ($\sim 0.6 \mu\text{m} - 30 \mu\text{m}$, $20 \text{ms} - 100 \text{s}$), well above and below the range that typical implementations of SPT and PIV can measure^{23,69}, particularly for dense or noisy systems. Moreover, our dynamical characterization of the surrounding network—which quantifies macromolecular dynamics, deformations and relaxation profiles in response to strain—is distinct and arguably much richer than mapping a simple strain or displacement field that PIV can generate. While PIV can determine the displacement averaged over a portion of the imaged FOV and over a given lag time, it is relatively insensitive to the random thermal motion of molecules. However, how macromolecules respond to random thermal forces (e.g., whether and to what extent the diffusive motion is anomalous) can provide important insight into the nature of crowded and entangled systems. Therefore, we use DDM to not only determine the degree to which the motion of the macromolecules align with the strain, but to also quantify the random thermal fluctuations.

'We rationalize our results as arising from the varying propensity for polymers to build up at the strain edges versus align and entropically stretch along the strain path, likely dictated by the coupled effects of polymer entanglements, threadings, and the varying relaxation rates and mechanisms accessible to flexible DNA versus rigid microtubules. Importantly, our observations and interpretations of the intriguing physical phenomena that we present here require the direct coupling of macromolecular dynamics and deformation fields to stress response—made uniquely possible by coupling OpT microrheology with DDM.'

'Further, wherever possible, we position our discussion of physical mechanisms driving our results in the context of general polymer physics and rheology theory, such that our results can be generalized to other polymeric systems and users can draw from our arguments to make sense of their own results on other polymeric systems. Finally, it is important to note that DNA has been used as a model polymer system for decades now to elucidate open questions in polymer physics^{26,89-92}. As such, our results using DNA are generally transferrable to polymer systems with similar number of entanglements per chain. Notably,

our systems do include rings, and our results suggest that threading plays an important role in the dynamics, stress response and propagation. For example, more pervasive threading in the 0.65:0 system as compared to the other two systems, is the likely mechanism that drives increased confinement and subdiffusion at lengthscales below that of the entanglement tube confinement, as well as strain-aligned deformation and stretching that, in turn, facilitates shear-thinning in response to the local disturbance. These important findings are directly applicable to investigations of ring-linear blends which continue to be the topic of fervent research and debate^{33,35,36,38,76,86,87,93-97}.

'The intriguing dynamics described in the previous sections suggest distinct differences in the mechanical response of the systems. For example, the insensitivity of the 0.5:6.4 composite to strain rate is indicative of a more elastic-like response compared to the DNA blends. Further, the resonant dynamics shown in Figs 2,4,5 may imply a non-monotonic dependence of the stress response on strain rate that would be strongest for the 0.65:0 system and weakest for 0.5:6.4.'

Importantly, we note that while the trends we observe for δ and β are generally consistent with one another and corroborate our interpretations of the data, there are key differences that highlight the need to analyze both metrics to fully capture the dynamics. One important example can be seen by comparing the 0.65:0 data in Fig 5a with that of Fig 4e. Near the strain (small y , red data), β_1 is the most superdiffusive while the corresponding δ is the smallest (closer to 1) for this system compared to 0.5:0 and 0.5:6.4. While the former observation indicates the most pronounced strain-induced directed transport and deformation, the latter indicates the least degree of active transport (or most pronounced confinement) among the three systems – seeming contradictions. Moreover, far from the strain, δ values for the 0.65:0 system are lower ($\delta \approx 0.5$) and exhibit much weaker dependence on q than the other two systems. Taken together, these features suggest that the 0.65:0 system experiences the greatest degree of confinement across all lengthscales (above and below d_T) and y distances from the strain, a feature that is not readily obvious in our analysis of $\tau(q)$ and β_1 and β_2 .

We recognize these differences as indicators of the increased threading probability in 0.65:0 compared to the other systems, in particular the 0.5:6.4 system. As we describe above, threading contributes to the dynamics at all measured lengthscales, whereas entanglements only contribute at $\lambda > d_T$, suggesting that the subdiffusion and confinement that we measure for the 0.65:0 system, in particular, is dominated by threading events. In support of this physical picture, we previously showed that actin-microtubule composites exhibited δ values of ~0.5 to 0.7 with the lowest and highest values arising in composites that are highly crosslinked and entangled (no crosslinks), respectively⁶⁸. The observation that the quiescent (large y) δ values for the 0.65:0 system are closer to those reported for crosslinked systems, which have extremely slow relaxation timescales compared to entangled systems, suggests that threaded rings, which also relax much more slowly than entangled rings, play a principal role in the dynamics. This fact, coupled with the comparatively lower δ values near the strain, indicate that the strain-aligned superdiffusion we measure is likely due to entropic stretching and deformation of confined polymers, rather than center-of-mass strain-aligned motion. Notably, this important distinction requires analysis of both $\beta(q)$ and $\delta(q)$.'

'Now, considering the dynamics near the strain path, if the superdiffusive dynamics we observe in Fig 3 are indeed due to polymer motion, deformation, and stretching along the strain path, then the strain-rate dependence of β_1 should mirror that of the alignment factor A_F , which, as we discuss above, quantifies the degree to which the polymer motion and orientation are aligned with the strain path.'

'These different factors, which all serve to reduce threading events, support our conjecture that threading plays a principle role in the dynamics and deformation field near the strain by facilitating alignment and flow. Moreover, reduced threading in the DNA-MT composite is further corroborated by

less pronounced small- λ subdiffusion (smaller β_2) compared to the DNA blends (Fig 3f) which we understand to be principally mediated by threading, as described above.'

'The takeaway is that at lengthscales smaller than the tube diameter, threading dominates the dynamics for all systems, while entanglements dominate at larger lengthscales. The onset of entanglement dynamics enhances subdiffusion far from the strain and alignment near the strain, with the 0.65:0 blend exhibiting the strongest threading and entanglement effects.'

REVIEWERS' COMMENTS

Reviewer #2 (Remarks to the Author):

Thank you to the authors for carefully addressing all of my comments in the previous review. The authors have adequately responded to all of my comments.

Reviewer #3 (Remarks to the Author):

The revised version of the manuscript is still very long, dense, and overwhelming in terms of content. However, the authors went a long way toward addressing the concerns of the three reviewers, in particular by making their storytelling less flat and easier to follow, and by removing any ambiguous message. I am thus marginally in favor of publication of the revised version in Nature Communications, mainly because I see novelty and interdisciplinary value in combining optical tweezers and DDM, as the authors demonstrate here for the first time, especially for systems that are very challenging to study with more traditional approaches, such as particle tracking or fluorescence correlation spectroscopy.

NCOMMS-22-09238A Response to Reviewer Comments

Reviewer comments are reproduced in green font

Reviewer #2 (Remarks to the Author):

Thank you to the authors for carefully addressing all of my comments in the previous review. The authors have adequately responded to all of my comments.

We are pleased that the Reviewer appreciates our extensive revisions and finds our revised manuscript suitable for publication.

Reviewer #3 (Remarks to the Author):

The revised version of the manuscript is still very long, dense, and overwhelming in terms of content. However, the authors went a long way toward addressing the concerns of the three reviewers, in particular by making their storytelling less flat and easier to follow, and by removing any ambiguous message. I am thus marginally in favor of publication of the revised version in Nature Communications, mainly because I see novelty and interdisciplinary value in combining optical tweezers and DDM, as the authors demonstrate here for the first time, especially for systems that are very challenging to study with more traditional approaches, such as particle tracking or fluorescence correlation spectroscopy.

We are pleased that the Reviewer is in favor of publication and recognizes that our extensive revisions have addressed the Reviewers' concerns. To address the Reviewer's comment that our manuscript remains 'long' and 'dense', we have shortened and simplified many descriptions of our results in the manuscript. We have also moved our Methods section and discussion of theoretically predicted length and timescales to the SI, as Section S1 and Section S2.